# Platinum single-atom catalyst coupled with transition metal/metal oxide heterostructure for accelerating alkaline hydrogen evolution reaction

Kai Ling Zhou[1,2], Zelin Wang[1,2], Chang Bao Han[1✉], Xiaoxing Ke[1✉], Changhao Wang [1], Yuhong Jin[1], Qianqian Zhang[1], Jingbing Liu[1], Hao Wang[1✉] & Hui Yan[1]

Single-atom catalysts provide an effective approach to reduce the amount of precious metals meanwhile maintain their catalytic activity. However, the sluggish activity of the catalysts for alkaline water dissociation has hampered advances in highly efficient hydrogen production. Herein, we develop a single-atom platinum immobilized NiO/Ni heterostructure ($Pt_{SA}$-NiO/Ni) as an alkaline hydrogen evolution catalyst. It is found that Pt single atom coupled with NiO/Ni heterostructure enables the tunable binding abilities of hydroxyl ions (OH*) and hydrogen (H*), which efficiently tailors the water dissociation energy and promotes the H* conversion for accelerating alkaline hydrogen evolution reaction. A further enhancement is achieved by constructing $Pt_{SA}$-NiO/Ni nanosheets on Ag nanowires to form a hierarchical three-dimensional morphology. Consequently, the fabricated $Pt_{SA}$-NiO/Ni catalyst displays high alkaline hydrogen evolution performances with a quite high mass activity of 20.6 A mg$^{-1}$ for Pt at the overpotential of 100 mV, significantly outperforming the reported catalysts.

[1] Faculty of Materials and Manufacturing, Beijing University of Technology, Beijing, P. R. China. [2] These authors contributed equally: Kai Ling Zhou, Zelin Wang. ✉email: cbhan@bjut.edu; kexiaoxing@bjut.edu.cn; haowang@bjut.edu.cn

Hydrogen (H$_2$) has been regarded as the most promising energy carrier alternative to fossil fuels due to the environmental friendliness nature and high gravimetric energy density[1,2]. Electrocatalytic water splitting powered by wind energy or solar technologies for hydrogen generation is considered a sustainable strategy[3]. For an optimal electrocatalyst, minimizing the energy barrier and increasing the active sites are desirable for boosting the hydrogen evolution reaction (HER)[4–6]. Despite the significant progress that has been presented in non-precious catalysts[7,8], the platinum (Pt)-based materials are still regarded as the most active catalysts for HER due to its optimal binding ability with hydrogen[9–12]. However, the high cost and scarcity of Pt hamper its large-scale application in electrolyzers for H$_2$ production. Single-atom catalysts (SACs) provide an effective approach to reduce the amount of Pt meanwhile maintain its high intrinsic activity[13–16]. Recently, electrocatalytic HER in an alkaline condition has attracted more attention because catalyst systems are generally unstable in acidic media, resulting in safety and cost concerns in practice. Unfortunately, the alkaline HER activity of Pt-based catalysts is approximately two orders of magnitude lower than that in the acidic condition caused by the high activation energy of the water dissociation step[17–20]. Alkaline HER process involves two electrochemical reaction steps: (step (i)) electron-coupled H$_2$O dissociation to generate adsorbed hydrogen hydroxyl (OH*) and hydrogen (H*) (Volmer step), and (step (ii)) the concomitant interaction of dissociated H* into molecular H$_2$ (Heyrovsky or Tafel step)[21,22]. In particular, the additional energy in step (i) is required to overcome the barrier for splitting strong OH–H bond, leading to a hamper of Pt SACs for alkaline HER application. Therefore, reducing the water dissociation energy in Volmer step (step (i)) for Pt SAC in alkaline media becomes vital for large-scale H$_2$ production of industrialization.

Some strategies have been developed to improve Pt SACs HER activity. For instance, employing microenvironment engineering to immobilize single Pt atoms in MXene nanosheets (Mo$_2$TiC$_2$T$_x$) and onion-like carbon nanospheres supports could greatly reduce the H adsorption energy ($\Delta G_H$) and, thus, facilitates the release of H$_2$ molecular[23,24]. Besides, Pt single atoms anchored alloy catalysts (Pt/np-Co$_{0.85}$Se SAC) were constructed as an efficient HER electrocatalyst[25], in which np-Co$_{0.85}$Se can largely optimize the adsorption/desorption energy of hydrogen on atomic Pt sites,

thus improving the HER kinetics. Furthermore, by utilizing the electronic interaction between the Pt atoms and the supports, single-atom Pt-anchored 2D MoS$_2$ (Pt$_{SA}$-MoS$_2$)[26], nitrogen-doped graphene nanosheets (Pt$_{SA}$-NGNs)[27], and porous carbon matrix (Pt@PCM)[28] show enhanced electrocatalytic HER efficiency due to the higher $d$-band occupation near Fermi level, which can provide more free electrons for boosting the H* conversion. Despite significant progress in Pt SACs, these methods are difficult to decrease the energy barrier of water dissociation in the Volmer step (step (i)). Generally, the H$_2$O dissociation and H* conversion happen on different catalytic sites[29]. Especially, the HER activities of Pt-based catalysts in alkaline conditions are governed by the binding ability of hydroxyl species (OH*)[18,30,31], and the alkaline HER kinetics could be optimized by independently regulating the binding energy of reactants (OH and H*) on dual active sites[32–34]. Inspired by these findings, the energy barrier of Pt SCAs for H$_2$O dissociation in Volmer step (step (i)) in alkaline media could be decreased by incorporating or creating the dual active sites in the catalyst to independently modulate the binding energy of reactants (OH* and H*).

In this work, we developed a three-dimensional (3D) nanostructured electrocatalyst consisting of two-dimensional (2D) NiO/Ni heterostructure nanosheets supported single-atom Pt attached on one-dimensional Ag nanowires (Ag NWs) conductive network (Pt$_{SA}$-NiO/Ni). Density functional theory (DFT) calculations reveal that the dual active sites consisting of metallic Ni sites and O vacancies-modified NiO sites near the interfaces of NiO/Ni heterostructure in Pt$_{SA}$-NiO/Ni show the preferred adsorption affinity toward OH* and H*, respectively, which efficiently facilitates water adsorption and reaching a barrier-free water dissociation step with a lower energy barrier of 0.31 eV in Volmer step (step (i)) for Pt$_{SA}$-NiO/Ni in the alkaline condition compared with that of Pt$_{SA}$-Ni (0.47 eV) and Pt$_{SA}$-NiO (1.42 eV) catalysts. In addition, anchoring Pt single atoms at the interfaces of NiO/Ni heterostructure induces more free electrons on Pt sites due to the elevated occupation of the Pt 5$d$ orbital at Fermi level and the more suitable H binding energy ($\Delta G_{H*}$, −0.07 eV) than that of Pt atoms at the NiO ($\Delta G_{H*}$, 0.74 eV) and Ni ($\Delta G_{H*}$, −0.38 eV), which efficiently promotes the H* conversion and H$_2$ desorption, thus accelerating overall alkaline HER (step (ii)). Furthermore, the Ag NWs-supported 3D morphology provides

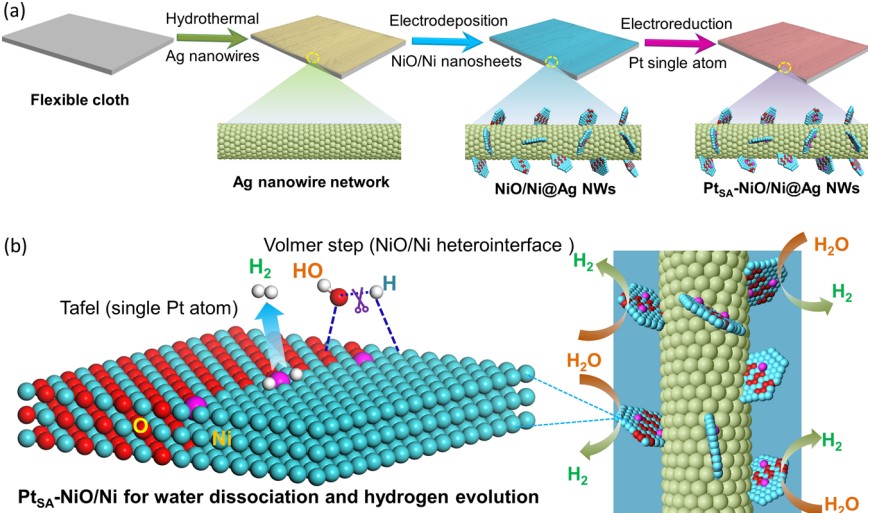

**Fig. 1 Schematic illustration of synthesis and water splitting mechanism of Pt$_{SA}$-NiO/Ni. a** The synthesis process of Pt single atom anchored NiO/Ni heterostructure nanosheets on Ag nanowires network. **b** The mechanism of Pt$_{SA}$-NiO/Ni network as an efficient catalyst toward large-scale water electrolysis in alkaline media.

abundant active sites and accessible channels for charge transfer and mass transport. As a result, the fabricated $Pt_{SA}$-NiO/Ni catalyst exhibits outstanding HER activity with a quite lower overpotential of 26 mV at 10 mA cm$^{-2}$ in 1-M KOH. The mass activity of $Pt_{SA}$-NiO/Ni is 20.6 A mg$^{-1}$ Pt at the overpotential of 100 mV, which is 41 times greater than that of the commercial Pt/C catalyst, significantly outperforming the reported catalysts. This work provides a design principle toward SAC systems for efficient alkaline HER.

## Results

**Synthesis and characterization of $Pt_{SA}$-NiO/Ni catalyst.** The fabrication process of $Pt_{SA}$-NiO/Ni on Ag NWs is illustrated in Fig. 1. In brief, the synthesized Ag NWs by a typical hydrothermal method[35] were first loaded on the flexible cloth to form a conductive network. The loading of Ag NWs leads to a brown film deposited on the surface of the white cloth fabric substrate (Fig. S1a, b), and the loading capacity of Ag NWs was determined to be ~0.47 mg cm$^{-2}$. The surface of the cloth fabric was studied by scanning electron microscopy (SEM) as shown in Fig. S1d–f, and a large number of fibers is presented. The abundant interconnected pores consist of a rich number of seams in each fiber. After the loading of the Ag NWs, the cloth fabric fibers are covered, and the uniform Ag NWs layer forms on the surface of cloth fabric as shown in Fig. S1g–i. Then Ni/NiO composite is attached to the Ag network by the facile electrodeposition process[36]. In detail, the Ag NWs network-loaded cloth is immersed in nickel acetate aqueous solution followed by an electrochemical process with −3.0 V versus SCE (saturated calomel electrode) for 200 s (Fig. S2), forming the uniformly distributed nanosheets on the Ag network (Fig. S3). Transmission electron microscopy (TEM, Fig. S4a, b) images, high-resolution TEM (HRTEM, Fig. S4c) image with corresponding fast Fourier transform (FFT pattern, Fig. S4d), and elemental mapping (Fig. S5) images clearly show that the metallic Ni is uniformly embedded in amorphous-like NiO nanosheets. Besides, the X-ray diffraction (XRD, Fig. S6) pattern shows that only metallic Ni signal without distinctive peaks of NiO can be detected, and X-ray photoelectron spectroscopy (XPS, Fig. S7) spectra suggest both metallic Ni and Ni oxide exists in Ni/NiO sample, further confirming the composition of metallic Ni on NiO. Interestingly, the deposited composition can be facilely controlled by performing various voltage in the nickel acetate aqueous solution[36]. Specifically, as above discussion, a high voltage of −3 V versus SCE will generate the Ni/NiO composite on Ag NWs (NiO/Ni), whereas a lower voltage of −1 V versus SCE could prepare the pure NiO on Ag NWs (NiO, Figs. S8–10). Besides, the pure metallic Ni on Ag network (Ni, Figs. S11–14) was fabricated by a traditional electrodeposition method with 1.2 V for 200 s in a mix solution containing 0.10-M NiCl$_2$ and 0.09-M H$_3$BO$_3$. Afterward, the single-atom Pt-immobilized NiO/Ni ($Pt_{SA}$-NiO/Ni) is obtained by sequentially electroreduction process with cyclic voltammetry in 1-M KOH solution containing low-concentration Pt metallic salts. Abundant voids and O vacancy defects at the surface-exposed interfaces of NiO/Ni heterostructure induced by crystal-lattice dislocation and phase transition[37–39] will provide efficient sites for trapping Pt single atom. The electrodeposition of $Pt_{SA}$-NiO/Ni leads to a black film deposited on the surface of Ag NWs@cloth fabric (Fig. S1b, c). In addition, the Ag NWs@cloth fabric supported $Pt_{SA}$-NiO/Ni catalyst also shown high wettability (Fig. S15). The water dissociation of Volmer step in alkaline aqueous media is expected to be accelerated by O vacancies-modified NiO near the interfaces interacted strongly with OH and metallic Ni interacted with H for H–OH bond destabilization (step (i)). Apart from the Volmer step, NiO/Ni

heterostructure-supported single-atom Pt sites could show more suitable H binding ability for the conversion and deabsorption of dissociated H (ii), further accelerating overall HER kinetics of $Pt_{SA}$-NiO/Ni in an alkaline condition.

The phase evolution of samples is investigated by XRD pattern as shown in Fig. 2a, in which no Pt characteristic peaks are detected, implying the absence of Pt cluster and particles in $Pt_{SA}$-NiO/Ni. The SEM (Fig. 2b, c) images show the well-distributed and open 3D nanosheets morphology for $Pt_{SA}$-NiO/Ni. During the single-atom Pt electroreduction process, some quantities of hydrogen bubbles are generated and released due to the high cathodic potential between 0 and −0.50 V versus reversible hydrogen electrode (RHE) in alkaline conditions[40]. In this case, the unchanged $Pt_{SA}$-NiO/Ni nanosheets morphology on Ag NWs compared with the original NiO/Ni (Fig. S3) indicates the high structural stability of the catalyst for HER application, and the exposed NiO/Ni nanosheet could also provide more sites for Pt atoms immobilization and improve the HER performance. The TEM (Fig. S16) images suggest that the nanosheets consist of few NiO/Ni layers for $Pt_{SA}$-NiO/Ni. The high-angle annular dark-field STEM (HAADF-STEM, Fig. 2d) image displays bright spots along with the interfaces of NiO/Ni heterostructure, corresponding to heavy constituent atoms species, which efficiently confirms the immobilization of atomically dispersed Pt atoms in the NiO/Ni nanosheets. The magnified HAADF-STEM image (Fig. 2e) suggests that the single Pt atoms are mostly immobilized at the interfaces of the NiO/Ni heterostructure. Based on these findings, the atomic environment of Pt atom was explored via the DFT-optimized structure (Figs. 2f, g and S17), and the result suggests that the Pt atoms are fixed at the Ni positions by binding with O atom and Ni atoms near the interfaces of the NiO/Ni heterostructure. Here, it needs to note that the theoretical prediction is limited due to the use of the crystalline NiO model instead of amorphous-like NiO during DFT calculation. Further, the HRTEM shows one distinct lattice fringes of 0.18 nm, matching well with metallic Ni (200) crystallographic planes (Fig. 2h). The FFT pattern (inset in Fig. 2h) shows four distinct rings: the red ring corresponds to the metallic Ni (200) plane[41], and the yellow rings with the highly diffused halo are assigned to the NiO phase[36,42]. These results further confirm the formation of single-atom Pt-anchored NiO/Ni composition, and the interfacial coupling of Pt single atom with NiO/Ni does not change the phase structure of NiO/Ni. Moreover, the elemental mapping, SEM image, and HAADF-STEM image (Figs. 2i–n and S18–20) show that Pt atoms are uniformly dispersed throughout NiO/Ni nanosheets. Besides, as a comparison, $Pt_{SA}$-NiO and $Pt_{SA}$-Ni were fabricated under the same conditions as $Pt_{SA}$-NiO/Ni but replacing NiO/Ni with NiO and Ni, respectively. The corresponding HAADF-STEM images (Fig. S21) confirm the atomically dispersed Pt in the NiO and metallic Ni phase.

The electronic state evolution of the single Pt atoms in NiO/Ni, NiO, and Ni supports is explored by XPS as shown in Fig. 3a. The Pt 4$f$ spectrums of $Pt_{SA}$-NiO/Ni, $Pt_{SA}$-NiO, and $Pt_{SA}$-Ni are close to Pt$^0$ but show some positive shift with different extents compared with Pt foil, confirming the electrochemical reduction of PtCl$_6^{2-}$ and the electronic interaction by charge transfer from Pt sites to the supports (NiO/Ni, NiO, and Ni)[43]. Specifically, the $Pt_{SA}$-NiO shows the largest positive shift in Pt 4$f$ spectrum, suggesting the maximum electron loss of Pt species[44,45]. Besides, the fitting curve of Pt XPS spectrums display Pt(IV) species in the samples, which derives from the adsorbed PtCl$_6^{2-}$ ions on the surface of the sample[46,47]. Further, the electronic state and atomic environment of Pt atoms in NiO/Ni, NiO, and Ni supports are further verified by performing X-ray absorption fine structure measurements. As shown in Fig. 3b, the evolutions of Pt $L_3$-edge X-ray absorption near edge structure (XANES) spectra with

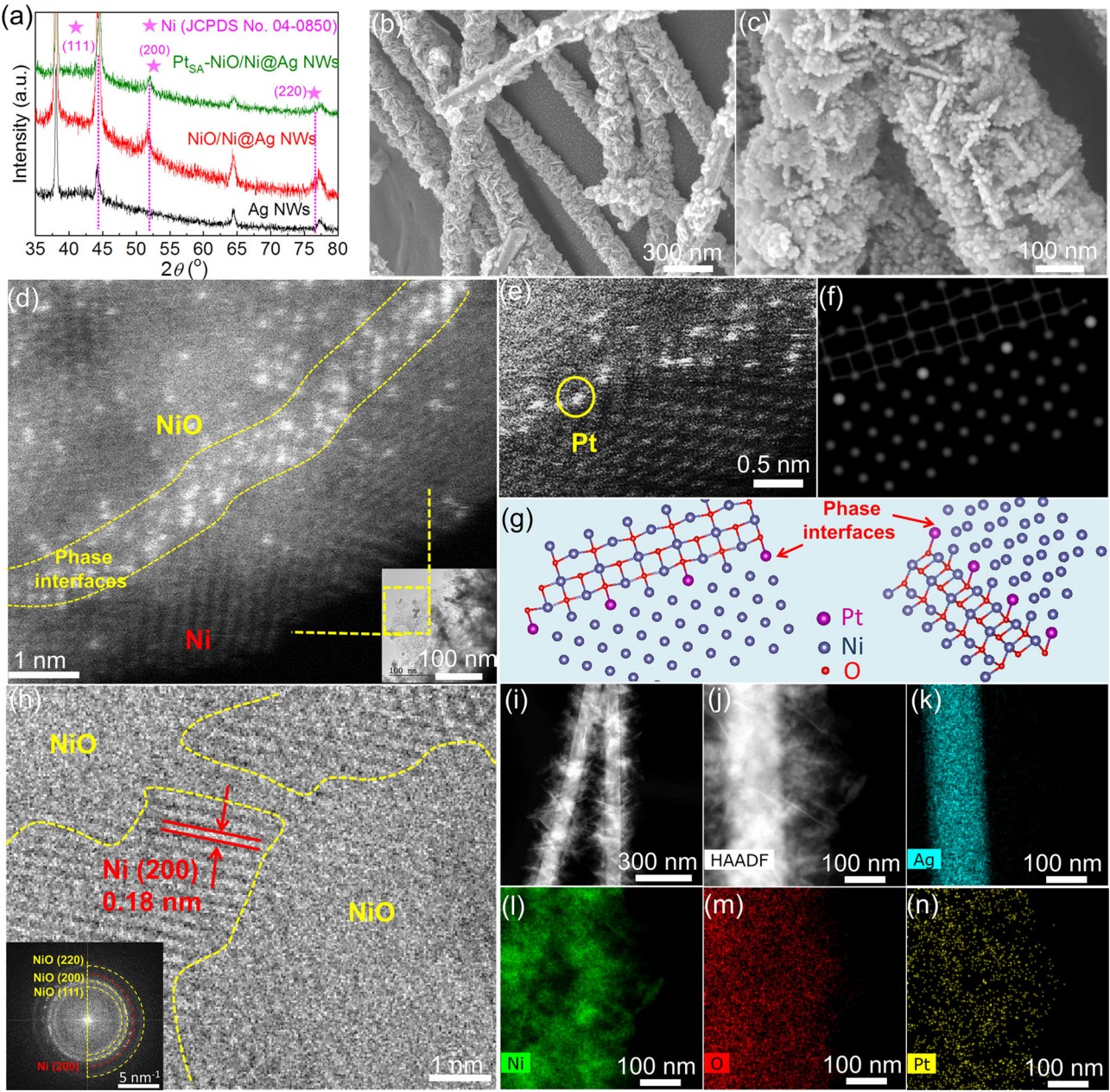

**Fig. 2 Structural characterization of the fabricated Pt_SA-NiO/Ni catalyst. a** XRD patterns of Pt_SA-NiO/Ni, NiO/Ni, and Ag NWs. **b, c** SEM images of Pt_SA-NiO/Ni. **d** HAADF-STEM image of Pt_SA-NiO/Ni. **e** Magnified HAADF-STEM image of Pt_SA-NiO/Ni and **f, g** the illustrated interface structure by DFT calculation, showing the atomically dispersed Pt atoms at Ni position (circles in (**e**)). **h** HRTEM images of Pt_SA-NiO/Ni and the insert in (**h**) show the related FFT pattern of Pt_SA-NiO/Ni. **i, j** HAADF-STEM images of Pt_SA-NiO/Ni at different magnifications and **k–n** the elemental mapping of the corresponding elementals.

different supports are distinguished, in which the intensity of white-line peaks corresponds to the transfer of the Pt $2p_{3/2}$ core-electron to $5d$ states, and thus is used as an indicator of Pt $5d$-band occupancy[27,48]. The overall white-line intensity gradually decreases as the change of support from NiO, NiO/Ni to metallic Ni, corresponding to the increase of $5d$ occupancy of Pt. Hence, higher $5d$ occupancy indicates the less charge loss of the single-atom Pt after coordinating with the supports, which is consistent with the results of XPS analysis in Fig. 3a.

To quantitate the structural information of the electronic state, the white-line peak evolution of Pt can be clearly described by the differential XANES spectra (ΔXANES, Fig. S22) by subtracting the spectra from that of Pt foil. The valence state of Pt can be quantitatively examined by the integration of the white-line peak in

ΔXANES spectra. As shown in Fig. 3c, the average valence state of Pt increase from +0.29, +0.73, to +1.23 for the Pt_SA-Ni, Pt_SA-NiO/Ni, and Pt_SA-NiO catalysts, respectively. The evolution of the atomic coordination configuration of Pt was further revealed by extended X-ray absorption fine structure spectroscopy (EXAFS, Fig. 3d), in which the typical Pt–Pt contribution peak of Pt foil at about 2.7 Å is absent for the fabricated Pt_SA-NiO/Ni, Pt_SA-NiO, and Pt_SA-Ni catalysts, strongly confirming the single Pt atoms dispersion. Further, the first-shell EXAFS fitting of Pt_SA-NiO/Ni sample (Fig. 3e and Table S1) gives a coordination number (CN) of 1.3 for Pt–O contribution and 5.8 for Pt–Ni contribution. For Pt_SA-NiO, the fitting results of EXAFS spectra suggested CN about 2.4 for Pt–O contributions and 2.1 for CN for Pt–Ni contributions. Whereas Pt–Ni contribution with 4.9 for CN and no Pt–O

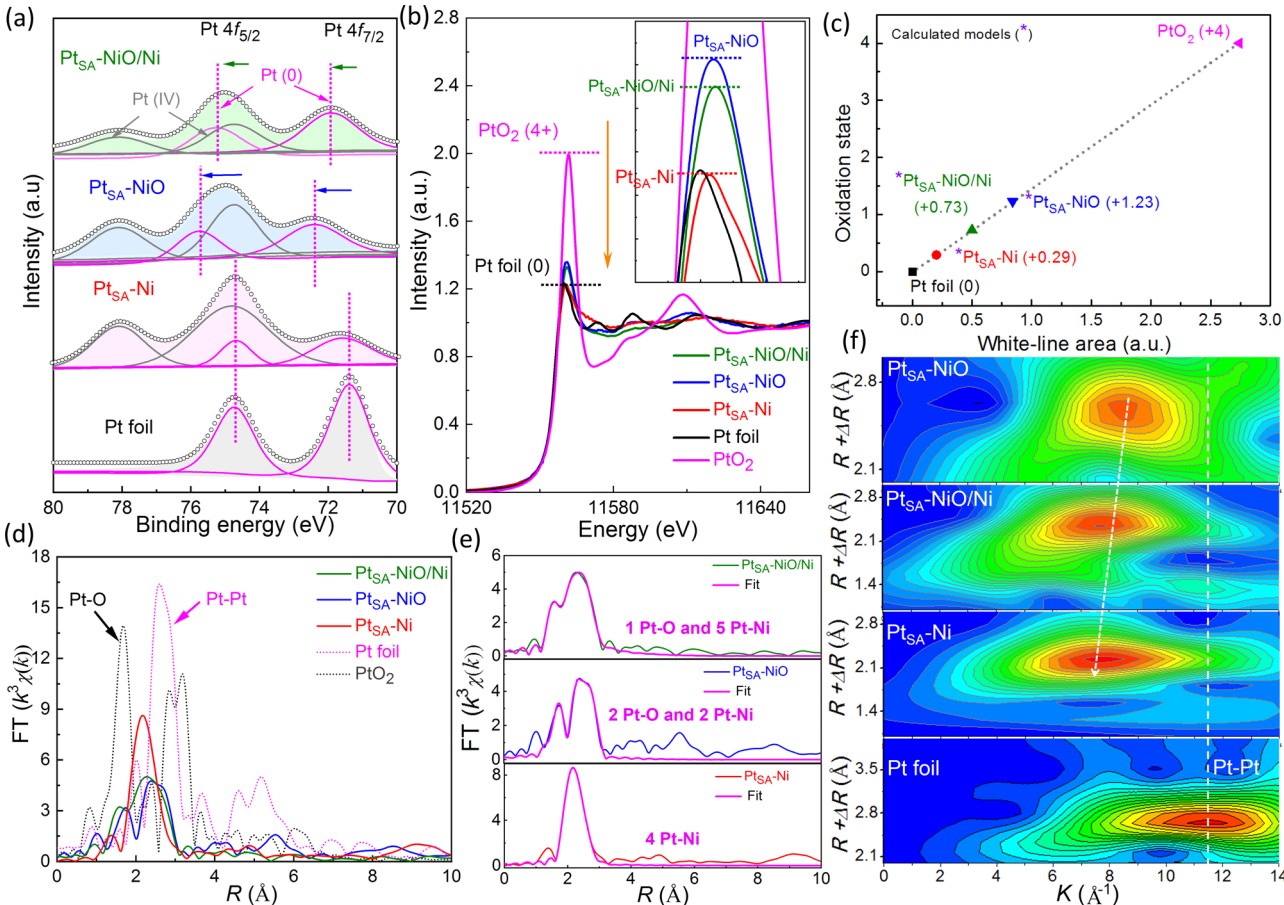

**Fig. 3 Electronic state and atomic structure characterization. a** Pt 4*f* spectra, **b** XANES spectra, and **c** calculated Pt oxidation states derived from ΔXANES spectra of Pt$_{SA}$-NiO/Ni, Pt$_{SA}$-NiO, and Pt$_{SA}$-Ni, and Pt foil is given as a reference. **d** Corresponding FT-EXAFS curves of Fig. 3b. **e** EXAFS fitting curve of Pt$_{SA}$-NiO/Ni, Pt$_{SA}$-NiO, and Pt$_{SA}$-Ni *R*-space. **f** EXAFS wavelet transform plots of Pt$_{SA}$-NiO/Ni, Pt$_{SA}$-NiO, Pt$_{SA}$-Ni, and Pt foil.

contributions are found in the fitting of Pt$_{SA}$-Ni EXAFS spectra. Combining the DFT-optimized structure (Fig. S23), the Pt atoms are mainly immobilized at the interfacial Ni positions by coordinating with one O atom and five Ni atoms in Pt$_{SA}$-NiO/Ni, which is consistent with the conclusion of HAADF-STEM analysis (Fig. 2d–g). To more precisely clarify the atomic dispersion and coordination conditions of Pt, the wavelet transform analysis was carried out due to its more efficient resolution ability in *K* spaces and radial distance[49,50], in which the atoms at similar coordination conditions and distances could be discriminated[51,52]. As shown in Fig. 3f, Pt$_{SA}$-NiO/Ni displays a different intensity maximum with Pt$_{SA}$-NiO and Pt$_{SA}$-Ni, and especially, the intensity maximum at 7.6 Å$^{-1}$ for Pt$_{SA}$-NiO/Ni is lower than that of Pt$_{SA}$-NiO (8.5 Å$^{-1}$), but high than that of Pt$_{SA}$-Ni (7.4 Å$^{-1}$), further confirming the interfacial coordination conditions for Pt atoms immobilized in NiO/Ni. Besides, the intensity maximum at 11.5 Å$^{-1}$ correspondings to Pt–Pt coordination is absent in the fabricated catalysts; further confirming the successful loading of single Pt atoms in Ni, NiO/Ni, and NiO supports.

**Theoretical investigations.** Based on the above structure analysis, theoretical investigations were performed to disclose the influences of the evolved coordinate configurations of the Pt atom on the electronic structure and catalytic activity of the catalysts. According to the HAADF-STEM and EXAFS measurements, the models for Pt$_{SA}$-NiO/Ni were shown in Fig. 4a. Based on the calculated charge density distributions, an increased charge density area along the interface of NiO/Ni heterostructure was induced (Fig. S24a, b). After coupling Pt single atom with NiO/Ni

heterostructure, an electronic structure redistribution at the interfaces of the heterostructure is caused due to the different electronegativity of atoms (3.44 for O atom, 1.91 for Ni, and 2.28 for Pt). Especially, charge delocalizing from Pt to the bonded O atom and charge localizing from adjacent Ni atoms to Pt are displayed. Consequently, a locally enhanced electric field with a half-moon shape area around the

Pt site was generated (Fig. S24c, d), which is more intensive than that of Pt$_{SA}$-NiO (Fig. 4b) and Pt$_{SA}$-Ni (Fig. 4c), suggesting Pt single atom coupled with NiO/Ni heterostructure could possess the more free electrons to promote the adsorbed H conversion and H$_2$ evolution[24,46]. Moreover, the projected density of states (PDOS, Figs. 4d and S25) of the single-atom Pt-immobilized NiO/Ni heterostructure shows higher occupation than that of the pure NiO/Ni heterostructure near the Fermi level, suggesting a promoted electron transfer and higher conductivity of Pt$_{SA}$-NiO/Ni. The contrast between the PDOS of NiO/Ni and Pt$_{SA}$-NiO/Ni reveals that the increased DOS of the Pt$_{SA}$-NiO/Ni near the Fermi level mainly derives from the contribution of Pt *d* orbitals (Fig. 4d). These results suggest that the NiO/Ni heterostructure-coupled single-atom Pt can effectively enhance the total *d*-electron domination of the catalyst near the Fermi level, which will benefit the activation of H$_2$O and lead to energetically catalytic activity[23,53]. Moreover, the *d*-band features of the Pt atom in NiO/Ni, NiO, and Ni coordinated configurations are investigated. The wider 5*d* band and higher density near the Fermi level for NiO/Ni-supported Pt atom than that of Pt$_{SA}$-NiO, and Pt$_{SA}$-Ni (Figs. 4e and S26) suggest that the NiO/Ni-coupled Pt atom can induce more free electrons near Pt sites than Pt$_{SA}$-NiO and Pt$_{SA}$-Ni, which is more favorable for the H reactants adsorption and

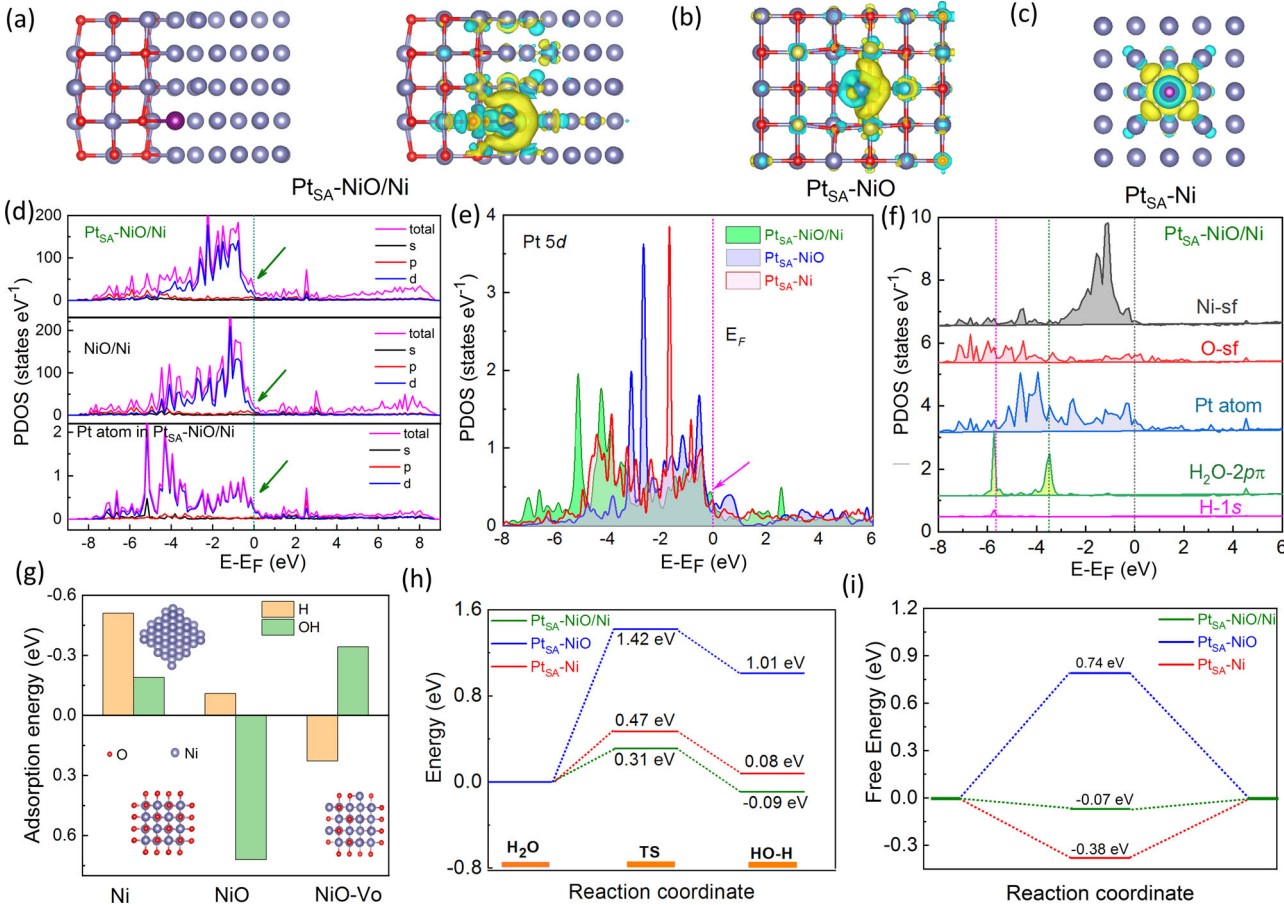

**Fig. 4 Theoretical investigations.** Computational models and localized electric field distribution of **a** $Pt_{SA}$-NiO/Ni, **b** $Pt_{SA}$-NiO, and **c** $Pt_{SA}$-Ni. **d** Calculated PDOS of NiO/Ni and $Pt_{SA}$-NiO/Ni, with aligned Fermi level. **e** Calculated Pt 5$d$ band of $Pt_{SA}$-NiO/Ni, $Pt_{SA}$-NiO, and $Pt_{SA}$-Ni. **f** The orbital alignment of the surficial sites for $Pt_{SA}$-NiO/Ni binding with $H_2O$ molecule. **g** Calculated OH-binding energies ($\Delta E_{OH}$) and H binding energies ($\Delta E_H$) for Ni, pure NiO, and O vacancies-modified NiO surface. **h** Calculated energy barriers of water dissociation kinetic and **i** adsorption free energies of H* on the surface of the $Pt_{SA}$-NiO/Ni, $Pt_{SA}$-NiO, and $Pt_{SA}$-Ni catalysts, respectively.

transfer. Besides, the Pt-5$d$ band of $Pt_{SA}$-NiO/Ni also shows a substantially broad range for overlapping with H-1$s$ and $H_2O$-2$p\pi$ orbitals (Fig. 4f). Therefore, the Pt site could play a protecting role for stabilizing the Ni valence state and a distributary role by binding OH and H species to low the deactivation of absorption sites in case of over-binding of intermediates on the active sites for NiO/Ni heterostructure-coupled single-atom Pt[54].

Based on the above finding, we further explore the reaction barrier of the fabricated catalysts for $H_2O$ splitting in alkaline conditions, consisting of the dissociation of $H_2O$ molecule of Volmer step and the subsequent conversion of H to $H_2$, which mainly depends on how OH and H bond to the active sites on the surface of the catalysts[55]. We found that both H and OH bind weakly to the pure NiO surface, and metallic Ni surface shows a preference for stabilizing H (Figs. 4g and S27). While O vacancies-modified NiO facilitates the adsorption of OH species (Figs. 4g and S28). For NiO/Ni composition, the O vacancies on the interfaces of the NiO/Ni heterostructure (Fig. S29) are induced by the crystal-lattice dislocation and phase transition[37–39]. As an integration, NiO/Ni-coupled single-atom Pt catalyst demonstrates the strongest $H_2O$ adsorption ability (Fig. S30) and largest energy release of −0.09 eV for water dissociation in Volmer step (Fig. 4h). Moreover, $Pt_{SA}$-NiO/Ni hybrid catalyst only needs the minimum energy barriers (0.31 eV) for the dissociation of $H_2O$ into OH and H under the assistance of NiO/Ni interfaces (Fig. S31) calculated by using the Ab Initio Cluster-Continuum Model, confirming the

critical role of surface-exposed NiO/Ni interfaces for the $H_2O$ dissociation of Volmer step in alkaline media. In the subsequent step, the NiO/Ni-supported single-atom Pt sites at the NiO/Ni interfaces act as the proton-acceptor for the recombination of the dissociated proton (H*) and $H_2$ evolution due to its near-zero H binding energy (−0.07 eV, Figs. 4i and S32) and strong electron supply capacity deriving from locally enhanced charge distribution (Fig. 4a) and the higher occupation of Pt 5$d$ band near Fermi lever (Fig. 4e). Consequently, the overall steps of $Pt_{SA}$-NiO/Ni hybrid catalyst for HER in alkaline media are significantly accelerated. Further, the effects of implicit solvation were considered by using VASPsol software as shown in Fig. S33, and NiO/Ni-coupled single-atom Pt catalyst also demonstrates the minimum energy barriers for the dissociation of $H_2O$ into OH and H than that of NiO-coupled single-atom Pt and Ni-coupled single-atom Pt catalyst (Fig. S33a), confirming the critical role of surface-exposed NiO/Ni interfaces for the $H_2O$ dissociation of Volmer step. Moreover, compared with $Pt_{SA}$-NiO and $Pt_{SA}$-Ni systems, the NiO/Ni-supported single-atom Pt sites at the NiO/Ni interfaces also show near-zero H binding energy (Fig. S33b), which is consistent with the results of Fig. 4h, i.

**Electrocatalytic alkaline HER performances.** Based on the structural characterizations and theoretical investigations, the Pt SAC-coupled with NiO/Ni heterostructure possesses the best

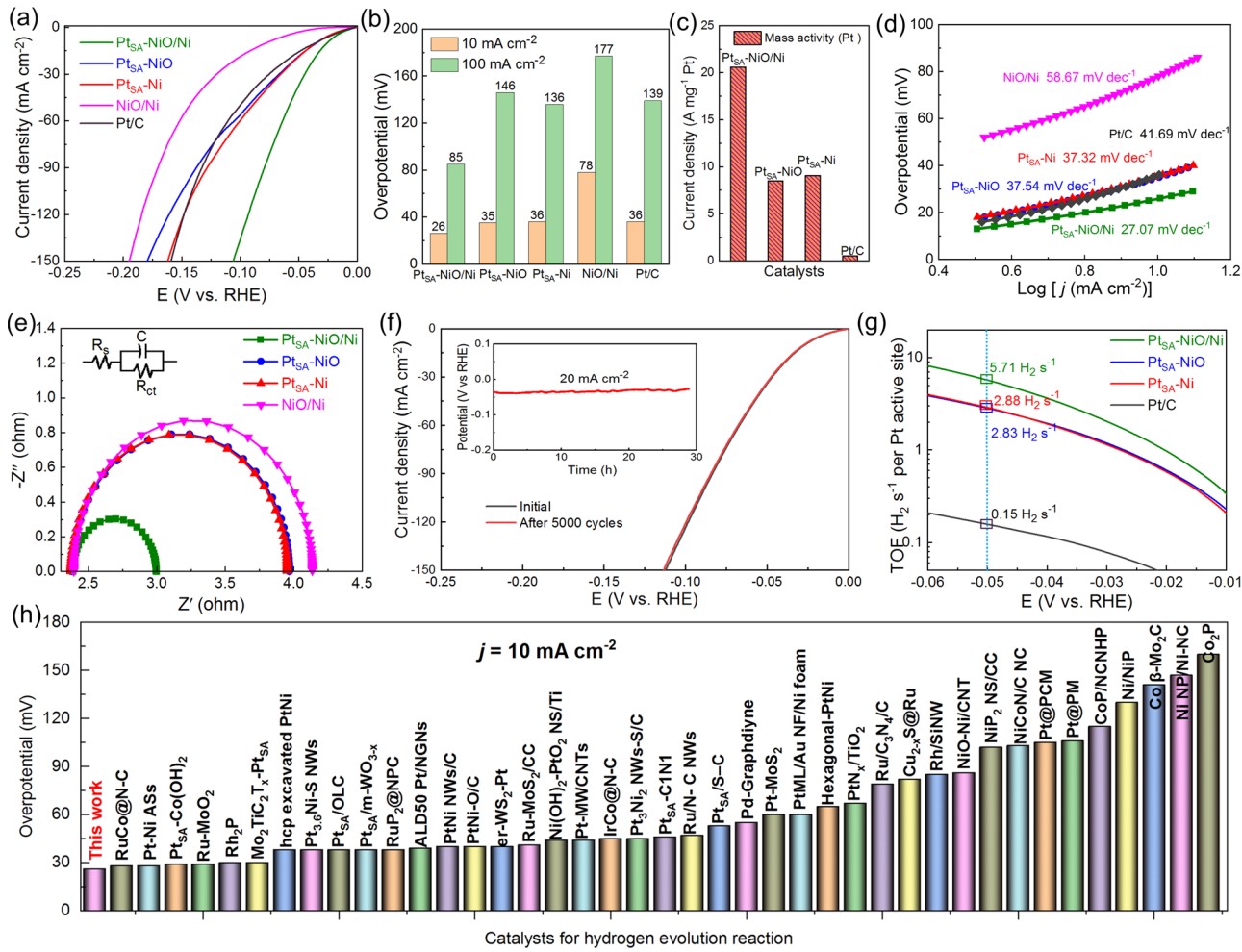

**Fig. 5 Electrocatalytic alkaline HER performances of the catalysts in 1-M KOH electrolyte. a** HER polarization curves of $Pt_{SA}$-NiO/Ni, $Pt_{SA}$-NiO, $Pt_{SA}$-Ni, NiO/Ni, and Pt/C. **b** The comparison of overpotentials required to achieve 10 and 100 mA cm$^{-2}$ for various catalysts. **c** The mass activity of the Pt-based catalysts. **d** Corresponding Tafel slope originated from LSV curves. **e** EIS (Electrochemical Impedance Spectroscopy) Nyquist plots of the catalysts. **f** Stability test of $Pt_{SA}$-NiO/Ni through cyclic potential scanning and chronoamperometry method (inset in **f**). **g** TOFs plots of the Pt-based electrocatalysts. **h** Comparison of the HER activity for $Pt_{SA}$-NiO/Ni with reported catalysts, originating from Table S3.

intrinsic HER activity in alkaline media among the fabricated catalysts. Thus, the electrocatalytic activities of $Pt_{SA}$-NiO/Ni for alkaline HER were measured in 1-M KOH solution. As a comparison, the HER performance of $Pt_{SA}$-NiO, $Pt_{SA}$-Ni, NiO/Ni, and 20% Pt/C was also tested under the same conditions. As shown in Fig. 5, the $Pt_{SA}$-NiO/Ni shows the highest HER performance among all catalysts, and only needs a quite low overpotential of 26 and 85 mV to achieve the current density of 10 and 100 mA cm$^{-2}$, respectively, significantly superior to the $Pt_{SA}$-NiO, $Pt_{SA}$-Ni, NiO/Ni, and the Pt/C catalyst (Fig. 5b). Moreover, the mass activity of $Pt_{SA}$-NiO/Ni normalized to the loaded Pt mass (1.14 wt%, inductively coupled plasma-mass spectrometry) at an overpotential of 100 mV is 20.6 A mg$^{-1}$, which is 2.4, 2.3, and 41.2 times greater than that of $Pt_{SA}$-NiO (8.5 A mg$^{-1}$), $Pt_{SA}$-Ni (9.0 A mg$^{-1}$), and the commercial Pt/C catalyst (0.5 A mg$^{-1}$), respectively. These results suggest that single Pt atoms coupled with NiO/Ni can maximize the alkaline HER activity of Pt-based catalysts, leading to a significant reduction in cost. In addition, the $Pt_{SA}$-NiO/Ni exhibits a smaller Tafel slope of 27.07 mV dec$^{-1}$ than $Pt_{SA}$-NiO (37.54 mV dec$^{-1}$), $Pt_{SA}$-Ni (37.32 mV dec$^{-1}$), NiO/Ni (58.67 mV dec$^{-1}$), and Pt/C catalyst (41.69 mV dec$^{-1}$), which suggests a typical Volmer-Tafel mechanism for alkaline HER and implies that the rate-determining step of $Pt_{SA}$-NiO/Ni is the $H_2$ desorption (Tafel step) rather than the $H_2O$ dissociation

(Volmer step)[34,56]. Besides, $Pt_{SA}$-NiO/Ni catalyst exhibits a 2.0, and 2.4-fold enhancement in the double-layer capacitance ($C_{dl}$) over $Pt_{SA}$-NiO and $Pt_{SA}$-Ni (Fig. S34), respectively, suggesting the favorable nanostructure with more sites for Pt atoms immobilization and HER. Furthermore, the charge transfer resistance ($R_{ct}$) of $Pt_{SA}$-NiO/Ni (0.61 ohm, Fig. 5e) is low than that of $Pt_{SA}$-NiO, $Pt_{SA}$-Ni, and NiO/Ni catalysts, which mainly originates from the introduction of cloth fabric substrate and Ag NWs (Figs. S35 and 36) and the enhanced electronic structure of single Pt atoms coupled with NiO/Ni.

For real applications, HER catalyzing stability is another essential factor. As present in Fig. 5f, the $Pt_{SA}$-NiO/Ni shows high durability in the alkaline electrolyte with negligible loss in HER performance for 5000 cycles or 30 h. The characterizations of $Pt_{SA}$-NiO/Ni after the stability test, including HAADF-STEM images, elemental mapping, and double-layer capacitance (Figs. S37–39), suggest the negligible structure changes and single-atom dispersion for $Pt_{SA}$-NiO/Ni after long-term alkaline HER. Moreover, the turnover frequencies (TOFs) per Pt atom site are analyzed, and the TOFs of $Pt_{SA}$-NiO/Ni (5.71 $H_2$ s$^{-1}$) are 2.02, 1.99, and 38.06 times higher than that of $Pt_{SA}$-NiO, $Pt_{SA}$-Ni, and Pt/C catalyst, respectively (Fig. 5g). To our knowledge, the electrocatalytic HER performances of our $Pt_{SA}$-NiO/Ni catalyst in the alkaline media are almost optimal among the reported SACs,

and are comparable with the performances of catalysts in acid media (Fig. 5h and Table S3), confirming the advance by the constructing single Pt sites in NiO/Ni hybrid system.

Further, based on the highly intrinsic HER activity, the electrocatalytic HER performances of the catalysts in neutral electrolytes containing 1.0-M phosphate buffer solutions (pH = 7.0) are investigated as shown in Fig. S40. $Pt_{SA}$-NiO/Ni shows the highest HER performance among all catalysts in neutral electrolytes, and only needs a quite low overpotential of 27 and 159 mV to achieve the current density of 10 and 100 mA cm$^{-2}$, respectively, significantly superior to the $Pt_{SA}$-NiO, $Pt_{SA}$-Ni, NiO/Ni, and the Pt/C catalyst (Fig. S40b). Moreover, Fig. S40c presents a small Tafel slope (31.94 mV dec$^{-1}$) for $Pt_{SA}$-NiO/Ni, lower than that of $Pt_{SA}$-NiO (47.26 mV dec$^{-1}$), $Pt_{SA}$-Ni (40.68 mV dec$^{-1}$), and Pt/C catalyst (42.40 mV dec$^{-1}$), revealing fast HER kinetics for NiO/Ni heterostructure-coupled Pt single atoms. The above merits of the $Pt_{SA}$-NiO/Ni, including low overpotential and Tafel slope, are superior to most previously reported catalysts in the neutral solution (Fig. S40d and Table S4), further confirming the advance by the constructing single Pt sites in NiO/Ni hybrid system.

## Discussion

In summary, we reported a single-atom Pt ($Pt_{SA}$) immobilized NiO/Ni heterostructure nanosheets on Ag NWs network nanocomposite by the facile electrodeposition strategy, which serves as an efficient electrocatalyst for vigorous hydrogen production in alkaline media. Theoretical calculations revealed that the Pt SACs coupled with NiO/Ni heterostructure could efficiently tailoring water dissociation energy for accelerating alkaline HER. In particular, the dual active sites consisting of metallic Ni sites and O vacancies-modified NiO sites near the interfaces of NiO/Ni have the preferred adsorption affinity toward both OH* and H*, which facilitates water adsorption and reaches a barrier-free water dissociation step with the lowest energy barrier of 0.31 eV in Volmer step (step (i)) for $Pt_{SA}$-NiO/Ni compared with that of $Pt_{SA}$-Ni (0.47 eV) and $Pt_{SA}$-NiO (1.42 eV) catalysts. Besides, fixing Pt atoms at the NiO/Ni interfaces induce a higher occupation of the Pt 5d band at the Fermi level and the more suitable H binding energy ($\Delta G_{H*}$, −0.07 eV) than that of Pt atoms at the NiO ($\Delta G_{H*}$, 0.74 eV) and Ni ($\Delta G_{H*}$, −0.38 eV), which efficiently promotes the H* conversion and H$_2$ desorption, thus accelerating overall alkaline HER. The further enhancement of alkaline HER performance was achieved by introducing the Ag NWs network into 2D $Pt_{SA}$-NiO/Ni nanosheets to construct a seamlessly conductive 3D nanostructure. The unique nanostructural feature and highly conductive Ag NWs network provide abundant active sites and accessible channels for electron transfer and mass transport. Consequently, the 3D $Pt_{SA}$-NiO/Ni catalyst shows outstanding HER performances in alkaline conditions with a quite low overpotential of 26 mV at a current density of 10 mA cm$^{-2}$ and high mass activity of 20.6 A mg$^{-1}$ Pt in 1-M KOH, significantly outperforming the reported catalysts. This study opens an efficient avenue for the advance of SACs by introducing a water dissociation kinetic-oriented material system.

## Methods

**Synthesis of Ag NWs**. An oil bath method was used to synthesize Ag NWs according to our previous report[57]. Specifically, a mix solution consisting of ethylene glycol, FeCl$_3$ (7.19 mM), AgNO$_3$ (0.051 M), and polyvinylpyrrolidone (0.012 M) was heat and maintained under an oil bath pan with 110 °C for 12 h. After that, the generated precipitate was washed with acetone and alcohol to get the pure Ag NWs. Subsequently, the Ag NWs were uniformly dispersed on a flexible cloth fabric by spray coating technology to fabricate a conductive network.

**Synthesis of NiO/Ni on Ag NWs**. Ni/NiO is grown on Ag NWs network by a facile electrodeposition process in the aqueous electrolyte of 20-mM

$C_4H_6NiO_4·4H_2O$ according to the recent report[36]. The electrodeposition process was performed by chronoamperometry method with −3 V versus SCE for 200 s under a standard three-electrode system, in which graphite sheet acted as a counter electrode, SCE acted as a reference electrode, and the fabricated Ag NWs network loaded on the cloth was directly used as working electrode. The obtained samples were washed with deionized water and then dried at room temperature.

**Synthesis of NiO on Ag NWs**. NiO is grown on Ag NWs network by the electrodeposition process with −1 V versus SCE for 600 s in an aqueous electrolyte of 20-mM $C_4H_6NiO_4·4H_2O$. The obtained samples were washed with deionized water and then dried at room temperature.

**Synthesis of Ni on Ag NWs**. Metallic Ni is grown on Ag NWs network by the electrodeposition process in an aqueous solution consisting of 0.10-M NiCl$_2$, 0.09-M H$_3$BO$_3$, and a solvent containing ethanol and deionized water with 2:5 in volume ratio. The electrodeposition process was performed by chronoamperometry with −1.2 V versus SCE for 200 s. The obtained samples were washed with deionized water and then dried at room temperature.

**Synthesis of $Pt_{SA}$-NiO/Ni on Ag NWs**. $Pt_{SA}$-NiO/Ni on Ag NWs was fabricated by the electrochemical reduction process in the three-electrode system, in which the fabricated NiO/Ni on Ag NWs was performed as the working electrode, graphite sheet acted as a counter electrode, saturated calomel electrode acted as a reference electrode. The corresponding electrochemical process was carried out by multi-cycle cathode polarization in 1-M KOH solution containing 50-μM H$_2$PtCl$_6$ with a scan rate of 50 mV s$^{-1}$ between 0 and −0.50 V versus RHE for 200 cycles.

**Synthesis of $Pt_{SA}$-NiO on Ag NWs**. $Pt_{SA}$-NiO on Ag NWs was fabricated by multi-cycle cathode polarization in 1-M KOH solution containing 50-μM H$_2$PtCl$_6$ with a scan rate of 50 mV s$^{-1}$ between 0 and −0.50 V versus RHE for 200 cycles.

**Synthesis of $Pt_{SA}$-Ni on Ag NWs**. $Pt_{SA}$-Ni on Ag NWs was fabricated by multi-cycle cathode polarization in 1-M KOH solution containing 50-μM H$_2$PtCl$_6$ with a scan rate of 50 mV s$^{-1}$ between 0 and −0.50 V versus RHE for 200 cycles.

**Characterizations**. The morphology measurement of the synthesized catalysts was performed by SEM (GeminiSEM 300). HRTEM images, HAADF-STEM images, and STEM-EDS mapping images were obtained by an FEI Titan G$^2$ microscope equipped with an aberration corrector for probe-forming lens and a Bruker SuperX energy dispersive spectrometer operated at 300 kV. The Pt contents in the catalysts were measured by inductively coupled plasma optical emission spectrometry. The XPS spectra of elementals were tested by a surface analysis system (ESCA-LAB250Xi). The phase and crystal information were obtained by Cu Kα radiation in an X-ray diffractometer (XRD, Bruker, D8 Advance Davinci). The EXAFS measurement of the $Pt_{SA}$-NiO/Ni, $Pt_{SA}$-NiO, and $Pt_{SA}$-NiO/Ni at the Pt $L_3$-edge was performed at 1W1B station at the Beijing Synchrotron Radiation Facility. Data analysis and fitting were performed with Athena and Artemis in the Demeter package.

**Electrochemical measurements**. All electrochemical measurements were finished by an electrochemical workstation (CHI 660E) with a three-electrode configuration, in which fabricated catalysts were directly employed as the working electrode, graphite sheet acted as a counter electrode, saturated calomel electrode acted as a reference electrode. All the presented potential in this work was transferred to RHE according to an experimental method[53]. LSV with 95% iR-corrections were tested under the potential range from 0.05 to −0.5 V and the scan rate of 5 mV s$^{-1}$. EIS was obtained by a frequency range from 100 k to 0.1 Hz with an overpotential of 230 mV versus RHE. For the preparation of 3D Pt/C@Ni foam, 5 mg 20-wt% Pt/C was dispersed in 0.9-mL alcohol containing 0.1 mL 5-wt% Nafion solution to form a homogeneous ink. Then, the obtained ink was coated on the Ni foam and dried in air to form a porous Pt/C@Ni foam electrode.

**DFT theoretical calculations**. All the structural optimizations, charge density difference analysis, Bader charge analysis, and energy calculations were carried out based on DFT as implemented in the Vienna Ab initio Simulation Package[58–60]. The projector-augmented wave method was implemented to calculate the interaction between the ionic cores and valence electrons[61,62]. The Perdew–Burke–Ernzerhof approach of spin-polarized generalized gradient approximation was used to describe the exchange-correlation energy[63]. Calculations were performed with the cutoff plane-wave kinetic energy of 500 eV, and 8 × 4 × 1 k-mesh grids were employed for the integration of the Brillouin zone. Electronic relaxation was undertaken to utilize the conjugate-gradient method[64] with the total energy convergence criterion being 10$^{-5}$ eV. Geometry optimization was employed by the quasi-Newton algorithm[65,66] until all the residual forces on unconstrained atoms <0.01 eV/Å. Climbing image nudge elastic band calculations[67] were employed for finding transition barriers with the initial configuration of H$_2$O absorbed on the catalyst surface and final configuration of OH +

H absorbed on the catalyst surface. To ensure the initial configuration correctly, an $H_2O$ molecule was deposited on the catalyst surface and relaxed for calculating its local minimum total energy on different sites, and the last one is the initially stable configuration. The final configuration is also found by relaxing OH and H near the $H_2O$ absorbed site of the initial configuration. Next, The equation for calculating adsorption enthalpy $\Delta E_{H*}$ as the following:

$$\Delta E_{H*} = E_{slab+H} - E_{slab} - \frac{1}{2}E_{H_2} \qquad (1)$$

where the $E_{slab+H}$ is the total enthalpy of H adsorbing on the catalysts, the enthalpy of the catalysts is $E_{slab}$, and the $H_2$ enthalpy is $E_{H_2}$.

The H and $H_2O$ absorbing on the slabs were investigated by comparing the formation energy of different sites. The equation for calculating adsorption enthalpy $\Delta E_{H*}$ as the following:

$$\Delta E_{H*} = E_{slab+H} - E_{slab} - \frac{1}{2}E_{H_2} \qquad (2)$$

where the $E_{slab+H}$ is the total enthalpy of H adsorbing on the catalysts, enthalpy of the catalysts is $E_{slab}$, the $H_2$ enthalpy is $E_{H_2}$. As similar, the equation for calculating the $H_2O$ adsorption enthalpy $\Delta E_{H_2O*}$ as the following:

$$\Delta E_{H_2O*} = E_{slab+H_2O} - E_{slab} - E_{H_2O*} \qquad (3)$$

The free energy of adsorbed H and $H_2O$ as follows:

$$\Delta G_{H*} = \Delta E_{H*} + \Delta E_{ZPE} - T\Delta S \qquad (4)$$

$$\Delta G_{H_2O*} = \Delta E_{H_2O*} + \Delta E_{ZPE} - T\Delta S \qquad (5)$$

where $\Delta E_{H*}$ represent the H adsorption energy and $\Delta E_{H_2O*}$ represent the $H_2O$ adsorption energy, and $\Delta E_{ZPE}$ represents the difference related to the zero-point energy between the gas phase and the adsorbed state.

## Data availability

The data that support the findings of this work are available from the corresponding author upon reasonable request.

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

## Acknowledgements

This work was supported by the National Natural Science Foundation of China (NSFC) (Grant No. 52070006, 11804012, 12074017), the Scientific and Technological Development Project of the Beijing Education Committee (No. KZ201710005009), and the Beijing Municipal Education Commission (Grant No. KM201910005009), the Beijing municipal high level innovative team building program (IDHT20190503) and the National Natural Science Fund for Innovative Research Groups of China (51621003).

## Author contributions

H.W. and H.Y. supervised this study. K.L.Z. conceived the idea. K.L.Z., Z.W., C.W., and Y.J. planned and carried out the experiments, collected, and analyzed the experimental data. X.K. and Q.Z. performed SEM and TEM characterizations. C.W. and K.L.Z. conducted theoretical calculations. K.L.Z., J.L., and C.B.H. wrote the paper. All the authors have discussed the results and wrote the paper together.

## Competing interests

The authors declare no competing interests.
