## [Peer Review File · Nature Communications]

REVIEWER COMMENTS

Reviewer #1 (Remarks to the Author):

This work describes single Pt-atom coupled with NiO/Ni junction for HER in alkaline electrolytes. The major finding is that Pt single atom located at NiO/Ni junctions accelerate water dissociation and hydrogen desorption, thereby improve HER kinetics. The authors present a large set of experimental and theoretical results to support their claims. Although the results are interesting and could be good contributions to the field, I feel there are some critical problems which I will explain below, such that the experiments are not well designed and the conclusions are not well supported. In addition, this work is on catalyst material development and if this catalyst is truly exceptional, compelling results in the more challenging neutral electrolytes should be provided.

1. the use of flexible cloth. fabric is a big concern, no information about the surface properties and chemical structure is provided and it is unclear how this cloth would affect performance.
2. no information of how much Ag coating was used and why Ag nanowire is necessary
3. NiO/Ni junction is known to be good HER catalysts in alkaline electrolytes, I expect better results if the authors truly have NiO/Ni junction.
4. another big concern is why Pt single atom deposit on NiO/Ni, but not on Ag nanowire?
5. A better normalized HER results should be provided. the use of flexible cloth makes impossible to compare with literature results.

Reviewer #2 (Remarks to the Author):

The manuscript submitted by Zhou et al. re-ports the synthesis of Pt single atoms immobilized NiO/Ni heterostructure as an alkaline HER catalyst with high stability and small overpotential. They found that the metallic Ni sites and O vacancies modified NiO sites prefer the adsorption for both OH* and H*, which should be facilitated for water dissociation. Moreover, the Pt atoms fixed at the NiO/Ni interfaces could promote the H* conversion and H₂ desorption, thus accelerating over-all alkaline HER. The reported investigation is interesting and also expands the research area of alkaline HER to achieve better electrochemical performance. Both the experimental and calculated data are well explained. Therefore, the manuscript should be considered to be published after considering the comments below.

Comments:

1. The structure of the reported catalyst is quite complex, with single-atom Pt immobilized NiO/Ni heterostructure nanosheets on Ag NWs network. What is the prospect of this material for the real application?
2. I noticed that in the equivalent circuit of figure 5E, the authors provided one set of R_{ct} and C component, which suggest the catalyst exposes only one uniform catalyst-electrolyte interface. However, the single atom Pt catalysts are loaded on silver wire network, how to make sure the silver/silver oxide interface is fully covered with no exposure site? if not, two sets of C & R_{ct} representing silver/silver oxide and Pt single atom catalyst should be taken into consideration.
3. For the Pt 4f XPS results, the authors used symmetric peaks to fit the results. This is not quite correct because of a large number of electrons located near the Fermi level of Pt, and therefore, asymmetric line-shapes should be taken into consideration. This is reflected by the fact that even for Pt foil, the fitting results using symmetric peaks do not fit well with the experimental results. In addition, the peak area between 4f_{7/2} and 4f_{5/2} of the same species should be 4:3, which was neglected by the authors.
4. The authors mentioned that the main role of Ag NWs is to construct a conductive 3D nanostructure.

In that case, is it possible to change Ag NWs by other conductive NWs materials to reduce the cost and simplify the material fabrication process?

Reviewer #3 (Remarks to the Author):

The major claims of this paper are (1) the authors synthesized Pt SACs on a NiO/Ni heterostructure, (2) this HER catalyst is exceptionally efficient in alkaline media, and (3) the Pt/NiO/O interface provides "dual" active sites, which facilitate dissociative water adsorption. This work is novel and of interest to the community and wider field because it reports a new catalyst for alkaline HER that is competitive with the state-of-the-art. The experimental evidence is sufficient to justify claims 1 and 2, however, further theoretical evidence is required to justify claim 3 (see major and minor comments/questions below). With that being said, I think this paper will inspire new strategies for optimizing electrolyzers using SACs, heterostructures, and morphology. For these reasons, I recommend publication after the major and minor comments/questions below are addressed:

Major comments

1. Regarding Figure 4(a-c)

Previously, Norskov and coworkers predicted that the O and H coverage of the most stable Ni(111) surface in base depends on the potential (see Figures 4 and 5 in the reference below). The authors at least should calculate the H and OH adsorption energies for realistic O and H coverages on Ni (e.g., 1/4 monolayer) as a point of comparison. They also should calculate the H and OH adsorption energies for realistic O and OH coverages at Ni²⁺ on NiO as such passivation of under-coordinated, surface Ni²⁺ likely is preferred electrostatically.

[Norskov and coworkers 2008] <https://doi.org/10.1039/B803956A>

2. Regarding the exclusion of solvent effects

In this reviewer's opinion, it's no longer acceptable to publish computational catalysis investigations without at least testing the effects of implicit and explicit solvation. Since the authors are using VASP, they at least should recalculate their most important predictions using VASPsol. Additionally, they should examine the convergence of these predictions with respect to the number of explicit solvating water molecules (at the active site should suffice).

3. Modeling strongly correlated NiO with GGA-PBE (a semi-local exchange-correlation functional)

NiO is a Mott insulator due to electron-electron interactions, which are described poorly by DFT with semi-local exchange-correlation functionals like GGA-PBE. While it is not clear from Figure 4, I suspect that the authors predict NiO to be metallic when, at 0 K (to which static DFT calculations correspond), it has a band gap of 3-4 eV. With that being said, since the authors are trying to predict room-temperature properties, perhaps the use of GGA-PBE for NiO is less problematic. To benchmark the accuracy of their calculations for NiO, the authors should calculate a room-temperature property for which reference experimental data exists, e.g., the NiO formation free energy.

Minor comments/questions

1. "Despite the significant progress that has been presented in nonprecious catalysts, the HER performances are still second to platinum (Pt)-based materials due to its optimal binding ability with hydrogen.7-10" on page 2

While this may be the case in base, there are several excellent electrocatalysts for the HER in acid, e.g., the nickel phosphides.

<https://doi.org/10.1039/C4EE02940B>
<https://pubs.acs.org/doi/abs/10.1021/acscatal.7b04466>

2. "Transmission electron microscopy (TEM, Figure S3a-b) images, high-resolution TEM (HRTEM, Figure S3c) image, fast Fourier transform (FFT, Figure S3d), and elements mapping (Figure S4) images clearly show that the metallic Ni nanoparticles uniformly embed in amorphous NiO nanosheets." on page 5

Do the authors still observe the metallic Ni NPs after several electrocatalytic cycles?

3. "Compared with the original NiO/Ni (Figure S2), the exposed PtSA-NiO/Ni nanosheets morphology on Ag NWs should be attributed to the H₂-assisted delamination effect^{21,34} during Pt electro-reduction process in alkaline condition, which will provide more sites for Pt atoms immobilization and improve the HER performance." on page 6

I'm not sure I see the difference between Figure 2(b-c) and Figure S2 to which the authors are referring – can they please clarify?

4. "These results further confirm the formation of single-atom Pt anchored NiO/Ni composition, and the interfacial coupling of Pt single atom with NiO/Ni does not change the phase structure of NiO/Ni." on page 8

The authors should mention explicitly the fact that their theoretical predictions are limited by the fact that they use a crystalline model for amorphous NiO.

5. "Besides, the fitting curve of Pt XPS spectrums display Pt(IV) species in the samples, which derives from the adsorbed PtCl₆²⁻ ions on the surface of the sample.^{44,45}" on page 8

Pt(IV) also could derive from Pt(IV) oxide as proposed by Birss et al in 1986 and demonstrated by Favaro et al in 2017. Can the authors distinguish between these two potential sources of the Pt(IV) XPS signal?

[Birss et al 1986] <https://doi.org/10.1149/1.2108978>
[Favaro et al 2017] <https://doi.org/10.1039/C7TA00409E>

6. "Combining the DFT-optimized structure (Figure S18), the Pt atoms are mainly immobilized at the interfacial Ni positions by coordinating with one O atom and 5 Ni atoms in PtSA-NiO/Ni, which is consistent with the conclusion of HAADF-STEM analysis (Figure 2d-g)." on page 10

For PtSA-NiO/Ni, what about the configuration where Pt adsorbs at the bridging site between two surface O? The reported Pt-O CN of ~1.3 suggests that Pt could adsorb on both sides of the interface. The authors should consider this.

Dear Editor:

We are glad to receive the reviewers' comments from you about our manuscript entitled "**Tailoring Water Dissociation Energy by Platinum Single-Atom Catalyst Coupled with Transition Metal/metal Oxide Heterostructure for Accelerating Alkaline Hydrogen Evolution Reaction**" (NCOMMS-21-03305). We thank you for your concern about our work. Here, we have noticed that three reviewers have commented on our paper. They give positive comments on our work, and further structural characterization and performance tests are required. We give our sincere thanks to all of the reviewers for taking the time to read and comment on our work. We understand that the strict comments suggest the high responsibility of reviewers for journal publishers and scientific research. According to the all of comments and suggestions proposed by reviewers, we carefully think about the deficiency of this work and make some significant improvements, so that our work can be competent for publishing on *Nature Communications*.

The main corrections have been added into the revised manuscript with yellow background, and the responses to the reviewer's comments are as following:

Reviewer #1:

This work describes single Pt-atom coupled with NiO/Ni junction for HER in alkaline electrolytes. The major finding is that Pt single atom located at NiO/Ni junctions accelerate water dissociation and hydrogen desorption, thereby improve HER kinetics. The authors present a large set of experimental and theoretical results to support their claims. Although the results are interesting and could be good contributions to the field, I feel there are some critical problems which I will explain below, such that the experiments are not well designed and the conclusions are not well supported.

Q1: This work is on catalyst material development and if this catalyst is truly exceptional, compelling results in the more challenging neutral electrolytes should be provided.

A1: Thank you for your comments on our work. Your doubt and valuable advice help us improve the quantity of this manuscript. The electrocatalytic HER performances of

the catalysts in neutral electrolytes containing 1.0 M phosphate buffer solutions (PBS, pH = 7.0) have been added in **Figure S40** and **Table S4** in the revised manuscript and also shown as **Figure R1** in this letter.

Based on the structural characterizations and theoretical investigations, the Pt single-atom catalyst coupled with NiO/Ni heterostructure possesses extremely highly intrinsic HER activity. As shown in **Figure R1a-b**, the Pt_{SA}-NiO/Ni shows the highest HER performance among all catalysts in 1.0 M PBS neutral electrolytes, and only needs a quite low overpotential of 27 and 159 mV to achieve the current density of 10 and 100 mA cm⁻², respectively, significantly, suggesting the superior neutral HER performance over the Pt_{SA}-NiO, Pt_{SA}-Ni, NiO/Ni and the Pt/C catalyst. Moreover, **Figure R1c** presents a small Tafel slope (31.94 mV dec⁻¹) for Pt_{SA}-NiO/Ni, lower than that of Pt_{SA}-NiO (47.26 mV dec⁻¹), Pt_{SA}-Ni (40.68 mV dec⁻¹), and Pt/C catalyst (42.40 mV dec⁻¹), revealing fast HER kinetics for NiO/Ni heterostructure coupled Pt single atoms in neutral electrolytes. The above merits of the Pt_{SA}-NiO/Ni, including low overpotential and Tafel slope, are superior to most previously reported catalysts in the neutral solution (**Figure R1d** and **Table S4**), further confirming the advance by the constructing single-Pt sites in NiO/Ni hybrid system.

Figure R1. Electrocatalytic HER performances of the catalysts in 1 M PBS neutral electrolyte. (a) HER polarization curves of Pt_{SA}-NiO/Ni, Pt_{SA}-NiO, Pt_{SA}-Ni, and Pt/C. (b) The comparison of overpotentials required to achieve 10 mA cm⁻² for various catalysts. (c) Corresponding Tafel slope originated from LSV curves. (d) Comparison of the HER activity for Pt_{SA}-NiO/Ni with reported

catalysts, originating from Table S4.

Q2: The use of flexible cloth fabric is a big concern, no information about the surface properties and chemical structure is provided and it is unclear how this cloth would affect performance.

A2: Thanks for your valuable advice and the corresponding data have been added in the **Figure S1 and Figure S15** section in the revised manuscript and shown as **Figure R2-4** in this letter.

As shown in **Figure R2a-c**, the synthesized Ag nanowires (NWs) were subsequently loaded on the cloth fabric to form an electron-conductive network for the subsequent NiO/Ni electrodeposition and single-atom Pt anchoring. The loading of Ag NWs leads to a brown film deposited on the surface of the white cloth fabric substrate (**Figure R2a-b**), and the electrodeposition of Pt_{SA}-NiO/Ni leads to a black film deposited on the surface of Ag NWs@cloth fabric (**Figure R2b-c**). The surface of the cloth fabric was studied by scanning electron microscopy (SEM) as shown in **Figure R2d-f**, and a large number of fibers are presented. The abundant interconnected pores consist of a rich number of seams in each fiber. After the loading of the Ag NWs, the surface and seams of the cloth fabric fibers are uniformly covered by the Ag NWs layer as shown in **Figure R2g-i**. The cloth fabric was made from polyester fibers which have good elasticity, wrinkle resistance, shape retention, excellent wash-and-wear performance, and durability¹. In addition, polyester fibers have good resistance to strong alkalies and acids at room temperature and belong to an excellent insulator. The majority composition of polyester fibers is composed of terephthalic acid and ethylene glycol (PET) as shown in the insert in **Figure R2f**.² The favorable hydrogen release deriving from the interconnected pore structure and high stability originating from the good resistance to strong alkalies will contribute cloth fabric to be used as an excellent substrate for electrocatalytic hydrogen evolution in alkaline media.

Further, the wettability of the cloth fabric substrate supported catalyst was investigated by measuring the contact angle of the electrode. As shown in **Figure R3**,

it was difficult to measure the contact angle of the electrode as the water was absorbed by the felt instantaneously, indicating the super hydrophilic nature. The excellent hydrophilicity will boost the electrolyte accessibility, accelerate the mass transfer, reduce the charge transfer resistance of the electrode and increase the durability of the electrode.³

Figure R2. The digital images of (a) bare cloth fabric, (b) the Ag NWs loaded cloth fabric, and (c) the as-prepared Pt_{SA}-NiO/Ni@Ag NWs network on cloth fabric. The different-magnification SEM images of (d-f) the bare cloth fabric substrate and (g-i) the fabricated Pt_{SA}-NiO/Ni@Ag NWs on a cloth fabric substrate.

Figure R3. Contact angle measurement of the cloth fabric substrate supported Pt_{SA}-NiO/Ni@Ag NWs electrode.

Besides, the influence of the cloth fabric substrate on the HER performance of the electrocatalyst was investigated by loading Ag NWs on the different substrates, including steel sheet, PET, cloth fabric, and glassy carbon electrodes. As shown in **Figure R4**, the various substrates supported Ag NWs based electrodes show the negligible difference in HER performances before onset overpotential (potential required to reach the current density of -5 mA cm^{-2}), suggesting the negligible effect of the substrates on the intrinsic HER activity of the electrocatalyst. When the current density greater than -5 mA cm^{-2} , the cloth fabric substrate supported Ag NWs demonstrate slightly high HER performances comparing with other Ag NWs based electrodes, which should be attributed to the favorable mass transfer and hydrogen release derived from interconnected pore structure and the super hydrophilic nature of the cloth fabric substrate supported catalyst electrode as above discussion.

Figure R4. Electrocatalytic HER performances of the different substrates supported Ag NWs in 1 M KOH electrolyte, including steel sheet, PET, cloth fabric, and glassy carbon electrode supported Ag NWs.

Q3: No information of how much Ag coating was used and why Ag NWs are necessary.

A3: Thanks for your valuable advice and the corresponding data have been added in the **Figure S36** section in the revised manuscript and shown as **Figure R5** in this letter. The loading capacity of the Ag NWs was determined to be $\sim 0.47 \text{ mg cm}^{-2}$. The unique nanostructure feature with the large specific area and high electrons

conductivity contributes to Ag NWs outstanding advantages in highly efficient HER of electrocatalysts^{4,5}.

Herein, several catalytic systems with different current collectors were introduced as comparable groups. The Pt_{SA}-NiO/Ni@Ni foam, Pt_{SA}-NiO/Ni@Carbon cloth, and Pt_{SA}-NiO/Ni@Cu foam were prepared using the same procedure as the Pt_{SA}-NiO/Ni@Ag NWs except for choosing Ni foam, carbon cloth, and Cu foam as the current collector instead of Ag NWs, respectively. As shown in **Figure R5a**, the response current density of Pt_{SA}-NiO/Ni@Ag NWs was higher than other control groups at the same overpotential in the polarization curve, suggesting a superior electrocatalytic HER activity. The Pt_{SA}-NiO/Ni@Ag NWs catalytic system could deliver a current density of 10 mA cm⁻² at an overpotential of 27 mV, which is lower than that of the Pt_{SA}-NiO/Ni@Ni foam (46 mV), Pt_{SA}-NiO/Ni@Carbon cloth (40 mV), and Pt_{SA}-NiO/Ni@Cu foam (36 mV). Moreover, the mass activity of Pt_{SA}-NiO/Ni@Ag NWs normalized to the loaded Pt mass at an overpotential of 100 mV is 20.6 A mg⁻¹ (**Figure R5b**), which is about 2.1, 1.2, and 1.5 times greater than that of Pt_{SA}-NiO/Ni@Ni foam (9.9 A mg⁻¹), Pt_{SA}-NiO/Ni@Carbon cloth (16.7 A mg⁻¹) and Pt_{SA}-NiO/Ni@Cu foam (14.2 A mg⁻¹), respectively, suggesting that introducing Ag NWs into Pt_{SA}-NiO/Ni can extremely maximize the alkaline HER activity of Pt-based catalysts.

To get insight into the origin of the extraordinary HER performance of Ag NWs supported Pt_{SA}-NiO/Ni, the HER reaction kinetics of the fabricated Pt-based catalysts were measured by electrochemical impedance spectroscopy (EIS). As depicted in **Figure R5c**, Pt_{SA}-NiO/Ni@Ag NWs electrode exhibits a much lower R_{ct} value (0.61 Ω cm⁻²) than that of Pt_{SA}-NiO/Ni@Ni foam (1.19 Ω cm⁻²), Pt_{SA}-NiO/Ni@Carbon cloth (0.72 Ω cm⁻²), and Pt_{SA}-NiO/Ni@Cu foam (0.84 Ω cm⁻²), indicating Pt_{SA}-NiO/Ni@Ag NWs electrode holds a smaller interfacial charge-transfer resistance than the others. This is originated from the higher electronic conductivity of Ag NWs in the Pt_{SA}-NiO/Ni@Ag NWs electrode comparing with traditional Ni foam, carbon cloth, and Cu foam substrate,⁵ which greatly accelerate interfacial charge transfer and mass transfer⁶.

Furthermore, the electrochemical surface area (ECSA) of the Pt_{SA}-NiO/Ni attached to the different conductive electrodes was measured using the cyclic voltammetry technique to obtain the double-layer capacitance (C_{dl}), which was linearly proportional to the ECSA. As shown in **Figure R5d**, Pt_{SA}-NiO/Ni@Ag NWs electrode possessed a considerably high C_{dl} (9.0 mF cm⁻²) than Pt_{SA}-NiO/Ni@Ni foam (3.3 mF cm⁻²), Pt_{SA}-NiO/Ni@Carbon cloth (3.8 mF cm⁻²), and Pt_{SA}-NiO/Ni@Cu foam (3.6 mF cm⁻²), suggesting much more accessible active sites in Pt_{SA}-NiO/Ni@Ag NWs, which contributes to the superior HER performance of Pt_{SA}-NiO/Ni@Ag NWs over Pt_{SA}-NiO/Ni@Ni foam, Pt_{SA}-NiO/Ni@Carbon cloth, and Pt_{SA}-NiO/Ni@Cu foam. From the above discussion, the Ag NWs possess outstanding features in the design of electrocatalyst for the highly efficient hydrogen evolution due to its unique nanostructure feature and high electrons conductivity.

Figure R5. Electrocatalytic alkaline HER performances of the Pt_{SA}-NiO/Ni attached to the different conductive electrodes, including Ag NWs, Ni foam, carbon cloth, and Cu foam. (a) HER polarization curves, (b) the mass activity, (c) EIS (Electrochemical Impedance Spectroscopy)

Nyquist plots, and (d) the double-layer capacitance (C_{dl}) of the catalysts.

Q4: NiO/Ni junction is known to be good HER catalysts in alkaline electrolytes, I expect better results if the authors truly have NiO/Ni junction.

A4: Thank you for your comments, and your doubt help us improve the quantity of this manuscript. As shown in **Figure R6a**, high-resolution transmission electron microscopy image and fast Fourier transform (FFT) images clearly show that the metallic Ni uniformly embeds in NiO nanosheets, and the NiO/Ni junctions are generated. Here, the Ni/NiO junction is grown on the Ag NWs network (NiO/Ni@Ag NWs) by a facile electrodeposition process. As shown in **Figure R6b**, the corresponding alkaline HER activities were measured by a three-electrode system in 1 M KOH solution. Remarkably, the NiO/Ni@Ag NWs could deliver a current density of 20 and 150 mA cm⁻² at the low overpotential of 103 and 195 mV, respectively. NiO/Ni junction has been widely studied due to the good HER activity in alkaline electrolytes. Typically, Gong et al.⁷ reported the Ni/NiO junction attached to oxidized carbon nanotube (NiO/Ni-CNT) as shown in **Figure R6c**, which requires a high overpotential of about 120 mV to deliver a current density of 20 mA cm⁻² as shown in **Figure R6d**, which is inferior to the reported NiO/Ni@Ag NWs catalyst in this work. Recently, Li et al.⁸ also reported the Ni/NiO junction loaded on the carbon cloth (**Figure R6e**), and the Ni/NiO junction electrode needs an overpotential of about 300 mV to deliver the current density of 150 mA cm⁻² as shown in **Figure R6f**, which is extremely larger than that of the NiO/Ni@Ag NWs catalyst (at 195 mV for 150 mA cm⁻²). These results demonstrate that the NiO/Ni junction is truly generated in this work, and the superior HER activity of NiO/Ni@Ag NWs reported by this work over reported NiO/Ni junction not only originates from the NiO and Ni synergy effect but also derives from the unique nanostructure feature and high electrons conductivity of Ag NWs as above discussion.⁹

Figure R6. The fabricated NiO/Ni junction catalysts and the corresponding alkaline HER performances reported by (a-b) this study, (c-d) Gong et al.⁷ and (e-f) Li et al.⁸

Q5: Another big concern is why Pt single atom deposit on NiO/Ni, but not on Ag nanowire?

A5: Thank you so much for proposing this valuable suggestion, and the corresponding data have been added in **Figure S17**, **Figure S19**, and **Figure S20** section in the revised manuscript and shown as **Figure R7-11** in this letter.

Firstly, according to the fabrication sequence, the Ni/NiO composite layer is attached to the Ag network by the facile electrodeposition process. After that, Pt atoms are immobilized on NiO/Ni@Ag NWs by electroreduction process. When Ag NWs are compactly coated by NiO/Ni layer, the Pt atoms will be difficult to be immobilized on Ag NWs. Here, the protuberant NiO/Ni nanosheets on Ag NWs are stripped out by powerful ultrasonic treatment to verify if the Ag NWs is fully covered with no exposure site, and the cross-sections of Pt_{SA}-NiO/Ni@Ag NWs are obtained

by Focused Ion beam technique (FIB, FEI Helios Nanolab 600i FIB/SEM dual-beam system equipped with Energy Dispersive System detectors) as shown in **Figure R7**. Obviously, the Ag NWs are compactly coated by NiO/Ni layer, and no bare Ag surface could be observed. The Energy Dispersive System (EDS, **Figure R8a-b**) line profile of Ag and Ni element along the cross-section of Pt_{SA}-NiO/Ni@Ag NWs further proves the compact NiO/Ni coating layer wrapping Ag NWs. The high-angle annular dark-field scanning transmission electron microscopy (HAADF-STEM, **Figure R8c-d**) image displays a seamless junction between NiO/Ni and Ag NWs. The above results efficiently demonstrate the Ag NWs are compactly wrapped by the NiO/Ni layer, which could prevent the Ag from directly contacting and reacting with PtCl₆²⁻ ions in KOH solution during the depositing process of Pt atoms. Consequently, the Pt atoms are mainly deposited on NiO/Ni instead of Ag NWs.

Secondly, the above conclusion could be further verified by HAADF-STEM and the corresponding elemental mapping of the synthesized Pt_{SA}- NiO/Ni@Ag NWs as shown in **Figure R9**. A core-shell structure was clearly shown as discussed above. The “core” layer was composed of Ag elements (**Figure R9e**), and Ni, O, and Pt elements share the same area located at the “shell” layer (**Figure R9b-d**). The powerful ultrasonic treatment results in some damage and exfoliation of NiO/Ni shell layer as director of the red arrow in **Figure R9**. Interestingly, the evolution of Pt element distribution in the NiO/Ni shell area before and after ultrasonic treatment shows the same feature as that of Ni and O elements, further proving that the Pt atoms mainly deposit on NiO/Ni.

Figure R7. The exposed NiO/Ni and Ag NWs interfaces (a-b, d-e) and cross-section (c, f) of Pt_{SA}-NiO/Ni@Ag NWs obtained by powerful ultrasonic processing and Focused Ion beam (FIB) technique, and (g-i) the SEM images of bare Ag NWs as the references.

Figure R8. (a-b) The SEM image of the cross-section of Pt_{SA}-NiO/Ni@Ag NWs with the associated EDS line profile of Ag and Ni along the dotted white line. (c-d) The HAADF-STEM image of Pt_{SA}-NiO/Ni@Ag NWs.

Figure R9. HAADF-STEM image and EDS mapping of Pt_{SA}-NiO/Ni@Ag NWs after the powerful ultrasonic treatment.

Thirdly, according to the optimized crystal structures deriving from the DFT calculation (**Figure R10**), the formation energy of Pt immobilized at NiO/Ni is -2.10 eV, significantly lower than that of Pt immobilized at Ag NW (0.06 eV), suggesting the preference deposition of Pt atoms on NiO/Ni.

Fourthly, the Pt immobilized Ag NWs (Pt@Ag NWs) were prepared using the same procedure as Pt_{SA}-NiO/Ni@Ag NWs except for choosing Ag NWs instead of NiO/Ni@Ag NWs as an electrode. As shown in **Figure R11a-b**, the response current density of Pt_{SA}-NiO/Ni@Ag NWs was extremely higher than that of Pt@Ag NWs and NiO/Ni@Ag NWs at the same overpotential in the polarization curve. The superior electrocatalytic activity of Pt_{SA}-NiO/Ni@Ag NWs over Pt@Ag NWs directly confirms that Pt single atom mainly deposits on NiO/Ni rather than Ag NWs as above discussion.

Figure R10. The optimized structures and the formation energy of Pt immobilized at the (a) NiO/Ni and (b) Ag NW.

Figure R11. The alkaline HER performances of the fabricated Pt_{SA}-NiO/Ni@Ag NWs, Pt@Ag NWs, NiO/Ni@Ag NWs, and Ag NWs electrode.

Q6: A better normalized HER results should be provided. the use of flexible cloth makes it impossible to compare with literature results.

A6: Thanks for your thoughtful advice and the corresponding data have been added in

the **Figure S35** section in the revised manuscript and shown as **Figure R12** in this letter.

Glass carbon electrodes have been widely used in HER catalysts fabrication as a current collector.^{10,11} However, polymer binder (e.g., polytetrafluoroethylene and Nafion) and conductive agent are usually required to hold the active material (HER catalysts) on glass carbon electrode due to the weak binding ability between catalysts and planar glass carbon electrode. As a similar process, Pt_{SA}-NiO/Ni supported by Ag NWs loaded glass carbon electrode (Pt_{SA}-NiO/Ni@AgNWs@GCE) was fabricated by coating catalyst ink on the current collector using Nafion as an adhesive followed by an electrochemical procedure. In detail, 5 mg Ag NWs, 50 μ l Nafion (5 wt%), and 0.95 ml ethanol were mixed and then sonicated for 30 mins to form a dispersion solution. Then a part of the dispersion ink was dropped onto the glassy carbon electrode followed by drying at room temperature in a vacuum oven. The Ag NWs loading was determined to be ~ 0.47 mg cm⁻². Finally, Pt_{SA}-NiO/Ni@AgNWs@GCE was prepared using the same procedure as the Pt_{SA}-NiO/Ni@Ag NWs loaded cloth fabric substrate (Pt_{SA}-NiO/Ni@AgNWs@CFS) except for choosing Ag NWs loaded glass carbon electrode instead of Ag NWs loaded cloth fabric substrate. It is necessary to note that for the Ag NWs loaded flexible cloth fabric, the polymer binders are unnecessary due to the strong affinity ability between Ag NWs and cloth fibers.

The alkaline HER performances of Pt_{SA}-NiO/Ni@AgNWs@GCE were investigated under a standard three-electrode system. As shown in **Figure R12a-b**, the Pt_{SA}-NiO/Ni@AgNWs@GCE shows a negligible difference in HER performances with Pt_{SA}-NiO/Ni@AgNWs@CFS when the response current density is lower than -50 mA cm⁻². However, when the current density is greater than -50 mA cm⁻², the HER performances of Pt_{SA}-NiO/Ni@AgNWs@GCE demonstrate inferior to that of Pt_{SA}-NiO/Ni@AgNWs@CFS. To get insight into the origin of the extraordinary HER performance of Pt_{SA}-NiO/Ni@AgNWs@CFS, the reaction kinetics of the fabricated electrodes were measured by EIS. As depicted in **Figure R12c**, the Pt_{SA}-NiO/Ni@AgNWs@CFS exhibits a much low R_{ct} value (0.61 Ω cm⁻²) than

Pt_{SA}-NiO/Ni@AgNWs@GCE ($0.72 \Omega \text{ cm}^{-2}$), indicating the smaller interfacial charge-transfer resistance for Pt_{SA}-NiO/Ni@AgNWs@CFS. Furthermore, the ECSA of the different electrodes was measured using the cyclic voltammetry technique to obtain C_{dl} . As shown in **Figure R12d**, the Pt_{SA}-NiO/Ni@AgNWs@CFS possessed the higher C_{dl} (9.0 mF cm^{-2}) than Pt_{SA}-NiO/Ni@AgNWs@GCE (7.2 mF cm^{-2}), suggesting more accessible active sites in Pt_{SA}-NiO/Ni@AgNWs@CFS.

From the above discussion, the preparation of the traditional glass carbon electrode-loaded catalysts usually involves coating catalyst fines on the current collectors using adhesives such as Nafion. However, the incorporation of these insulating adhesives will inevitably bury active sites and increases the dead volume and the contact resistance between the catalyst and the current collector,¹² causing the insufficient utilization of active sites and poor electron transferability as shown in **Figure R12c-d**. Thus, integrating catalysts and current collectors to form a seamlessly conductive electrode is necessary. The extraordinary HER performances of Pt_{SA}-NiO/Ni@AgNWs@CFS are attributed to the high conductivity, more accessible active sites, the favorable mass transfer, and hydrogen release originating from the binder-free electrode fabrication process, interconnected pore structure, and the super hydrophilic nature of the cloth fabric substrate supported catalyst as above discussion in **Figure R2-3**. Even so, the HER performances of Pt_{SA}-NiO/Ni@Ag NWs loaded traditional glass carbon electrodes are superior to most previously reported catalysts as shown in **Figure R12e** due to the high intrinsic HER activity of single-Pt anchored NiO/Ni hybrid system.

Figure R12. The alkaline HER performances of the fabricated Pt_{SA}-NiO/Ni@Ag NWs loaded on the traditional glass carbon electrode and cloth fabric substrate. (a) HER polarization curves. (b) The comparison of overpotentials required to achieve 10 and 100 mA cm⁻² for various catalysts. (c) EIS Nyquist plots of the catalysts. (d) The corresponding scan rate dependence of the average currents. (e) Comparison of the HER activity for the traditional glass carbon electrode loaded Pt_{SA}-NiO/Ni@Ag NWs with reported catalysts, originating from Table S3.

Reviewer #2

The manuscript submitted by Zhou et al. re-ports the synthesis of Pt single atoms immobilized NiO/Ni heterostructure as an alkaline HER catalyst with high stability and small overpotential. They found that the metallic Ni sites and O vacancies modified NiO sites prefer the adsorption for both OH* and H*, which should be facilitated for water dissociation. Moreover, the Pt atoms fixed at the NiO/Ni interfaces could promote the H* conversion and H₂ desorption, thus accelerating overall alkaline HER. The reported investigation is interesting and also expands the research area of alkaline HER to achieve better electrochemical performance. Both

the experimental and calculated data are well explained. Therefore, the manuscript should be considered to be published after considering the comments below.

Q1: The structure of the reported catalyst is quite complex, with single-atom Pt immobilized NiO/Ni heterostructure nanosheets on the Ag NWs network. What is the prospect of this material for the real application?

A1: Thank you for your encouragement in our work. Your valuable advice helps us improve the quantity of this manuscript. The corresponding explanations have been added to the revised manuscript.

Firstly, the facile fabrication process: the fabrication of Pt_{SA}-NiO/Ni on Ag NWs only involves a two-step facile electrodeposition process as illustrated in **Figure R13**, in which Ni/NiO composite is attached to the Ag network by an electrochemical process with -3.0 V versus SCE (saturated calomel electrode) for 200 s, and then single-atom Pt is immobilized on NiO/Ni by sequentially electroreduction process with cyclic voltammetry for 200 cycles. Due to the fast and room temperature fabrication of the Pt_{SA}-NiO/Ni@Ag NWs electrode in an aqueous solution, the entire process is environmentally friendly, energy-efficient, affordable, and scalable to large-scale manufacturing.

Secondly, cost-effective catalyst: the mass activity normalized to the Pt loading for Pt_{SA}-NiO/Ni@Ag NWs is 20.6 A mg⁻¹ at the overpotential of 100 mV, which is 41 times greater than that of the commercial Pt/C catalyst, indicating that single-atom Pt anchored NiO/Ni@ could significantly maximize the electrocatalytic activity and reduce the cost of noble catalysts, suggesting a promising prospect of this material for the commercial application.

Thirdly, the indispensable role for outstanding HER performance: although the structure of Pt_{SA}-NiO/Ni@Ag NWs seems to be complex, every part of the structure is necessary and plays an indispensable role for the catalyst to display the superior HER performance over the reported SACs. Specifically, NiO/Ni heterostructure could efficiently lower the energy barrier of water dissociation of Volmer step in alkaline condition by utilizing the preferred adsorption affinity of the dual active sites consisting of metallic Ni sites and O vacancies modified NiO sites

near the interfaces of NiO/Ni in Pt_{SA}-NiO/Ni for H* and OH* groups, respectively. Additionally, the NiO/Ni tailored single-atom Pt is pivotal in inducing the more free electrons on Pt sites by improving the electrons occupation of Pt 5*d* band near Femi level and reaching a near-zero H binding energy (ΔG_{H^*} , -0.07 eV), which further promotes the H* conversion and H₂ evolution. Moreover, the introduction of Ag NWs could construct a hierarchical three-dimensional (3D) morphology that provides abundant active sites and accessible channels for charge transfer and mass transport. Consequently, the HER performance of Pt_{SA}-NiO/Ni@Ag NWs was extremely higher than that of Pt@Ag NWs, NiO/Ni@Ag NWs, Pt_{SA}-NiO@Ag NWs, Pt_{SA}-Ni@Ag NWs, the Pt/C catalyst, and the most previously reported catalysts as shown in **Figure R12e**, **Figure R11a-b** and **Figure 5**.

Figure R13. Schematic illustration of the synthesis process of Pt single atom anchored NiO/Ni heterostructure nanosheets on Ag NWs network.

Q2: I noticed that in the equivalent circuit of figure 5e, the authors provided one set of R_{ct} and C component, which suggest the catalyst exposes only one uniform catalyst-electrolyte interface. However, the single-atom Pt catalysts are loaded on a silver wire network, how to make sure the silver/silver oxide interface is fully covered with no exposure site? if not, two sets of C & R_{ct} representing silver/silver oxide and Pt single-atom catalyst should be taken into consideration.

A2: Thank you so much for proposing this valuable suggestion, and the corresponding data have been added in the **Figure S19** section in the revised manuscript and shown as **Figure R7-8** in this letter.

According to the fabrication sequence, the Ni/NiO composite layer is attached to the Ag network by the facile electrodeposition process. After that, Pt atoms are immobilized on NiO/Ni@Ag NWs by electroreduction process. When Ag NWs are compactly coated by NiO/Ni layer, the Ag NWs will be difficult to be directly in contact and react with the electrolyte. Here, the protuberant NiO/Ni nanosheets on Ag NWs are stripped out by powerful ultrasonic treatment to verify if the Ag NWs is fully covered with no exposure site, and the cross-sections of Pt_{SA}-NiO/Ni@Ag NWs are obtained by Focused Ion beam technique (FIB, FEI Helios Nanolab 600i FIB/SEM dual-beam system equipped with Energy Dispersive System detectors) as shown in **Figure R7**. Obviously, Ag NWs are compactly coated by NiO/Ni layer, and no bare Ag surface could be observed. The EDS (**Figure R8a-b**) line profile of Ag and Ni element along the cross-section of Pt_{SA}-NiO/Ni@Ag NWs further proves the compact NiO/Ni coating layer wrapping Ag NWs. HAADF-STEM (**Figure R8c-d**) image displays a seamless junction between NiO/Ni and Ag NWs. The above results efficiently demonstrate the Ag NWs are compactly wrapped by the NiO/Ni layer. So only Pt_{SA}-NiO/Ni interface is exposed in the electrolyte, which could prevent the Ag NWs from directly contacting and reacting with electrolyte during the HER process of Pt_{SA}-NiO/Ni@Ag NWs. So we use one set of R_{ct} and C components in the equivalent circuit of **Figure 5e**.

Q3: For the Pt 4f XPS results, the authors used symmetric peaks to fit the results. This is not quite correct because of the large number of electrons located near the Fermi level of Pt, and therefore, asymmetric line-shapes should be taken into consideration. This is reflected by the fact that even for Pt foil, the fitting results using symmetric peaks do not fit well with the experimental results. In addition, the peak area between 4f_{7/2} and 4f_{5/2} of the same species should be 4:3, which was neglected by the authors.

A3: Thanks for your professional advice and the corresponding data have been corrected and shown in **Figure 3a** in the revised manuscript and shown as **Figure R14** in this letter, in which the Pt XPS fitting curve of Pt_{SA}-NiO/Ni, Pt_{SA}-NiO, and Pt_{SA}-Ni samples is demonstrated by considering the asymmetric line-shapes and the

peak area ratio with 4:3 between $4f_{7/2}$ and $4f_{5/2}$.

As shown in **Figure R14**, the Pt $4f$ spectral of Pt_{SA}-NiO/Ni, Pt_{SA}-NiO, and Pt_{SA}-Ni are close to Pt⁰ but show some positive shift with different extents compared with Pt foil, confirming the electrochemical reduction of PtCl₆²⁻ and the electronic interaction by charge transfer from Pt sites to the supports (NiO/Ni, NiO, and Ni).¹¹ Specifically, the Pt_{SA}-NiO shows the largest positive shift in Pt $4f$ spectrum, suggesting the maximum electron loss of Pt species.^{13,14} Besides, the fitting curve of Pt XPS spectra display Pt(IV) species in the samples, which derives from the adsorbed PtCl₆²⁻ ions on the surface of the sample.^{4,15} These results are consistent with the results of Pt L_3 -edge X-ray absorption near edge structure (XANES) spectra in **Figure 3b** in the revised manuscript.

Figure R14. The corrected Pt XPS fitting curve of Pt_{SA}-NiO/Ni, Pt_{SA}-NiO, and Pt_{SA}-Ni samples by considering the asymmetric line-shapes and the peak area ratio between $4f_{7/2}$ and $4f_{5/2}$.

Q4: The authors mentioned that the main role of Ag NWs is to construct a conductive 3D nanostructure. In that case, is it possible to change Ag NWs by other conductive NWs materials to reduce the cost and simplify the material fabrication process?

A4: Thanks for your valuable advice and the corresponding data have been added in **Figure S36** section in the revised manuscript and shown as **Figure R5** in this letter.

From the above discussion, the Ag NWs possess outstanding features in the design of electrocatalyst for the highly efficient hydrogen evolution due to its unique nanostructure feature and high electrons conductivity.

Herein, under the consideration of the low cost, large-scale manufacturing, and the commercial application for Pt_{SA}-NiO/Ni-based catalyst, several catalytic systems with different conductive supporters were introduced investigated. The Pt_{SA}-NiO/Ni@Ni foam, Pt_{SA}-NiO/Ni@carbon cloth, and Pt_{SA}-NiO/Ni@Cu foam were prepared using the same procedure as the Pt_{SA}-NiO/Ni@Ag NWs except for choosing Ni foam, carbon cloth, and Cu foam as the conductive supporter, respectively. As shown in **Figure R5a**, the response current density of Pt_{SA}-NiO/Ni@Ag NWs was higher than other control groups at the same overpotential in the polarization curve. The Pt_{SA}-NiO/Ni@Ag NWs catalytic system could deliver a current density of 10 mA cm⁻² at an overpotential of 27 mV, which is lower than that of the Pt_{SA}-NiO/Ni@Ni foam (46 mV), Pt_{SA}-NiO/Ni@carbon cloth (40 mV), and Pt_{SA}-NiO/Ni@Cu foam (36 mV). Moreover, the mass activity of Pt_{SA}-NiO/Ni@Ag NWs normalized to the loaded Pt mass at an overpotential of 100 mV is 20.6 A mg⁻¹ as shown in **Figure R5b**, which is about 2.1, 1.2, and 1.5 times greater than that of Pt_{SA}-NiO/Ni@Ni foam (9.9 A mg⁻¹), Pt_{SA}-NiO/Ni@carbon cloth (16.7 A mg⁻¹) and Pt_{SA}-NiO/Ni@Cu foam (14.2 A mg⁻¹), respectively, suggesting that introducing Ag NWs into Pt_{SA}-NiO/Ni can extremely maximize the alkaline HER activity of Pt-based catalysts. By EIS and ECSA measurement shown in **Figure R5c-d**, the extraordinary HER performance of Ag NWs supported Pt_{SA}-NiO/Ni derive from the higher HER reaction kinetics (0.61 Ω cm⁻² for R_{ct}) and the larger electrochemical specific area (9.0 mF cm⁻² for C_{dl}) than others, confirming the unique nanostructure feature and high electrons conductivity of Ag NWs. Based on the above analysis, carbon cloth and Cu foam substrate serve as the potential substitute of Ag NWs under the consideration of the low cost, large-scale manufacturing, and the commercial application for Pt_{SA}-NiO/Ni-based catalyst, due to the superior HER performances for carbon cloth and Cu foam supported Pt_{SA}-NiO/Ni over most previously reported catalysts deriving from the highly intrinsic HER activity of single-Pt anchored NiO/Ni hybrid system as shown in **Figure R15**.

Figure R15. Comparison of the HER activity for carbon cloth and Cu foam supported Pt_{SA}-NiO/Ni with reported catalysts, originating from Table S3.

Reviewer #3:

The major claims of this paper are (1) the authors synthesized Pt SACs on a NiO/Ni heterostructure, (2) this HER catalyst is exceptionally efficient in alkaline media, and (3) the Pt/NiO/O interface provides "dual" active sites, which facilitate dissociative water adsorption. This work is novel and of interest to the community and wider field because it reports a new catalyst for alkaline HER that is competitive with the state-of-the-art. The experimental evidence is sufficient to justify claims 1 and 2, however, further theoretical evidence is required to justify claim 3 (see major and minor comments/questions below). With that being said, I think this paper will inspire new strategies for optimizing electrolyzers using SACs, heterostructures, and morphology. For these reasons, I recommend publication after the major and minor comments/questions below are addressed:

Q1: Regarding Figure 4(a-c): Previously, Norskov and coworkers [Norskov and coworkers 2008, <https://doi.org/10.1039/B803956A>] predicted that the O and H coverage of the most stable Ni(111) surface in base depends on the potential (see Figures 4 and 5 in the reference below). The authors at least should calculate the H and OH adsorption energies for realistic O and H coverages on Ni (e.g., 1/4 monolayer) as a point of comparison. They also should calculate the H and OH

adsorption energies for realistic O and OH coverages at Ni²⁺ on NiO as such passivation of under-coordinated, surface Ni²⁺ likely is preferred electrostatically.

A1: Thank you for your encouragement in our work. Your professional advice helps us improve the quantity of this manuscript. The corresponding data have been added in **Figure S27** section and **Table S2** in the revised manuscript and shown as **Figure R16** and **Table R1** in this letter.

According to your professional advice, the H and OH adsorption energies of Ni and NiO with the different coverages of 1/1, 3/4, 1/2, 1/4, and 1/8 are calculated, respectively, as shown in **Figure R16** and **Table R1**. Compared with the high coverage of H and OH on Ni and NiO, the lower coverages will lead to a stronger H and OH adsorption interaction, hinting that too crowded H or OH on Ni and NiO surfaces have the repulsion interaction with each other to make them hard to adsorb on the surface. Nevertheless, for the different coverages, both H and OH bind weakly to the pure NiO surface, while metallic Ni surface shows a preference for binding H, which is consistent with the analysis of **Figure 4g** in the original manuscript. These results suggest that for the different potential conditions, the binding preferences of H and OH on Ni and NiO surfaces are consistent. Herein, we use the adsorption energies of 1/4 coverages to update the values in **Figure 4g** in the revised manuscript.

Figure R16. The H and OH adsorption energies of (a-b) Ni and (c-d) NiO with the different coverages of 1/1, 3/4, 1/2, 1/4, and 1/8.

Table R1. H and OH adsorption energies on Ni (100) and NiO (100) surface with the different coverages of 1/1, 3/4, 1/2, 1/4, and 1/8, respectively. The unit is eV.

	1/1	3/4	1/2	1/4	1/8
Ni_H	-0.5253	-0.5219	-0.5186	-0.5177	-
Ni_OH	0.2252	0.0406	-0.2496	-0.1852	-
NiO_H	0.4506	0.2162	0.3986	-0.1123	-0.1221
NiO_OH	1.0381	0.8177	1.0001	0.7182	0.8234

Q2: Regarding the exclusion of solvent effects: In this reviewer's opinion, it's no longer acceptable to publish computational catalysis investigations without at least testing the effects of implicit and explicit solvation. Since the authors are using VASP, they at least should recalculate their most important predictions using VASPsol. Additionally, they should examine the convergence of these predictions concerning the number of explicit solvating water molecules (at the active site should be sufficed).

A2: Thanks for your professional advice and the corresponding data have been added

in **Figure 4h-i** and **Figure S33** section in the revised manuscript and shown as **Figure R17** in this letter.

The effects of implicit solvation were considered by using VASPsol software,¹⁶ and the revised energy barrier of water desorption and the adsorption free energies of H* for Pt_{SA}-NiO/Ni, Pt_{SA}-NiO, and Pt_{SA}-Ni systems were shown in **Figure R17a-b**. Under the effect of implicit solvation, NiO/Ni coupled single-atom Pt catalyst also demonstrates the minimum energy barriers (0.23 eV) for the dissociation of H₂O into OH and H than that of NiO coupled single-atom Pt and Ni coupled single-atom Pt catalyst (**Figure R17a**), confirming the critical role of surface-exposed NiO/Ni interfaces for the H₂O dissociation of Volmer step. Moreover, compared with Pt_{SA}-NiO and Pt_{SA}-Ni systems, the NiO/Ni supported single-atom Pt sites at the NiO/Ni interfaces also show near-zero H binding energy (-0.03 eV, **Figure R17b**), which is more favorable for the recombination of the dissociated proton (H*) and H₂ evolution.

For the effects of implicit solvation, we employed the Ab Initio Cluster-Continuum Model^{17,18} to reduce the requirement of the number of configurations and water molecules, which strategy converges solvation energies by surrounding an explicit water shell with the implicit solvent outside. The explicit water molecules were set around the catalytic active site of the model surface, and 100 configurations were obtained by the LAMMPS molecular dynamics software.¹⁹ After testing the explicit water molecules with the different numbers of 5, 14, 25, 35, 75, and 103, we found that the 35 water molecules could ensure the energy converged. The explicit water molecules revised energy barriers of water dissociation and the adsorption free energies of H* were shown in **Figure R17c-d**, respectively. The results exhibit the energy barriers of water dissociation with 0.31, 1.42, and 0.47 eV for Pt_{SA}-NiO/Ni, Pt_{SA}-NiO, and Pt_{SA}-Ni systems, respectively. The adsorption free energies of H* with -0.07, 0.74, and -0.38 eV for Pt_{SA}-NiO/Ni, Pt_{SA}-NiO, and Pt_{SA}-Ni systems, respectively, further confirming the critical role of surface-exposed NiO/Ni interfaces for the H₂O dissociation of Volmer step and NiO/Ni supported single-atom Pt for the recombination of the dissociated proton (H*) and H₂ evolution.

Figure R17. Calculated energy barriers of water dissociation kinetic and adsorption free energies of H^* on the surface of the $\text{Pt}_{\text{SA}}\text{-NiO/Ni}$, $\text{Pt}_{\text{SA}}\text{-NiO}$, and $\text{Pt}_{\text{SA}}\text{-Ni}$ catalysts by considering the effects of (a-b) implicit and (c-d) explicit solvation, respectively.

Q3: Modeling strongly correlated NiO with GGA-PBE (a semi-local exchange-correlation functional): NiO is a Mott insulator due to electron-electron interactions, which are described poorly by DFT with semi-local exchange-correlation functionals like GGA-PBE. While it is not clear from Figure 4, I suspect that the authors predict NiO to be metallic when, at 0 K (to which static DFT calculations correspond), it has a bandgap of 3-4 eV. With that being said, since the authors are trying to predict room-temperature properties, perhaps the use of GGA-PBE for NiO is less problematic. To benchmark the accuracy of their calculations for NiO, the authors should calculate a room-temperature property for which reference experimental data exists, e.g., the NiO formation free energy.

A3: Thanks for your professional advice and the corresponding data have been shown as **Figure R18-19** in this letter.

You are right in describing the NiO as the Mott insulator. In our DFT with the

exchange-correlation function of GGA-PBE, NiO demonstrates the metallic feature as shown in **Figure 18**.

In fact, in our DFT, the key factor affecting the catalytical speed is the energy barrier of water desorption (as shown in **Figure 4h**) and the adsorption free energies of H* (as shown in **Figure 4i**). So, the accuracy of energy is the key point in our DFT calculation. According to the reviewer's opinion, we calculate the room-temperature Helmholtz free energy $F(T, V)$ of NiO:

$$F(T, V) = E_0(V) + F_{vib}(T, V) + F_{el}(T, V)$$

where $E_0(V)$ is the ground-state total energy at an equilibrium volume V_0 , estimated by first-principles calculations; $F_{vib}(T, V)$ is the vibrational energy of the lattice ions. Under the quasi-harmonic approximation (QHA), the $F_{vib}(T, V)$ can be calculated by the equation:

$$F_{vib}(T, V) = k_B T \int_0^\infty g(\omega) \ln \left[\sinh \left(\frac{\hbar \omega}{2k_B T} \right) \right] d\omega$$

where ω is the phonon frequencies and $g(\omega)$ is the phonon density of states; while $F_{el}(T, V)$ is the thermal electronic contribution to the free energy, which can be expressed in terms of the electronic entropy $S_{ele}(T, V)$ and the electron excitation energy $E_{ele}(T, V)$ by the equation $F_{el}(T, V) = E_{ele}(T, V) + T * S_{ele}(T, V)$.

From the calculation result of Helmholtz free energy (**Figure R19**), we found the energy changes from -11.38 to -11.46 eV per formula unit with the temperature from 0 to 300 K. It proves the energy does not change too much (0.08 eV per formula unit). It should not affect our calculation accuracy apparently, because the energy barrier of water desorption and the adsorption free energies of H* only have a relationship with the interaction between the NiO slab and H₂O/H*, but they do not include the energy of the NiO slab.

Figure 18. The electronic density of states of pure NiO with bulk structure using static GGA + PBE at 0 K.

Figure 19. Helmholtz free energy versus temperature from 0 K to 300 K for NiO. The enthalpy unit is eV per formula unit.

Q4: "Despite the significant progress that has been presented in nonprecious catalysts, the HER performances are still second to platinum (Pt)-based materials due to its optimal binding ability with hydrogen.⁷⁻¹⁰" on page 2. While this may be the case in base, there are several excellent electrocatalysts for the HER in acid, e.g., the nickel phosphides (<https://doi.org/10.1039/C4EE02940B>, <https://pubs.acs.org/doi/abs/10.1021/acscatal.7b04466>)

A4: Thanks for your professional reminder. The corresponding references and correct description have been added as **references 7-8** in the revised manuscript and shown as **references 20-21** in this letter.

Despite the significant progress that has been presented in nonprecious catalysts,^{20,21} the platinum (Pt)-based materials are still regarded as the most active catalysts for HER due to their optimal binding ability with hydrogen.²²⁻²⁵

Q5: "Transmission electron microscopy (TEM, Figure S3a-b) images, high-resolution TEM (HRTEM, Figure S3c) image, fast Fourier transform (FFT, Figure S3d), and elemental mapping (Figure S4) images clearly show that the metallic Ni nanoparticles uniformly embed in amorphous NiO nanosheets." on page 5. Do the authors still observe the metallic Ni NPs after several electrocatalytic cycles?

A5: Thanks for pointing this out. The HER catalyzing stability of NiO/Ni@Ag NWs was investigated by carrying out the cyclic voltammetry with 200 cycles in 1 M KOH solution. After the cyclic voltammetry test, the metallic Ni nanoparticles could be still observed in the HRTEM image and related FFT pattern of NiO/Ni@Ag NWs shown in **Figure R20**, suggesting the high stability of NiO/Ni heterostructure.

Figure R20. (a) HRTEM image and (b) the related FFT pattern of NiO/Ni@Ag NWs after the cyclic voltammetry test with 200 cycles in 1 M KOH solution.

Q6: "Compared with the original NiO/Ni (Figure S2), the exposed Pt_{SA}-NiO/Ni nanosheets morphology on Ag NWs should be attributed to the H₂-assisted delamination effect^{21,34} during Pt electro-reduction process in alkaline condition, which will provide more sites for Pt atoms immobilization and improve the HER performance." on page 6. I'm not sure I see the difference between Figure 2(b-c) and

Figure S2 to which the authors are referring – can they please clarify?

A6: Thanks for pointing this out. The corresponding descriptions have been modified and clarified in the revised manuscript.

During the single-atom Pt electro-reduction process, some quantities of hydrogen bubbles are generated and released due to the high cathodic potentials between 0 V and -0.50 V versus reversible hydrogen electrode (RHE) in alkaline conditions.²⁶ In this case, the unchanged Pt_{SA}-NiO/Ni nanosheets morphology on Ag NWs (**Figure 2b-c**) compared with the original NiO/Ni (**Figure S3**) indicates the high structural stability of the catalyst for HER application, and the exposed NiO/Ni nanosheet could provide more sites for Pt atoms immobilization and improve the HER performance.

Q7: "These results further confirm the formation of single-atom Pt anchored NiO/Ni composition, and the interfacial coupling of Pt single atom with NiO/Ni does not change the phase structure of NiO/Ni." on page 8. The authors should mention explicitly the fact that their theoretical predictions are limited by the fact that they use a crystalline model for amorphous NiO.

A7: Thanks for pointing this out. Indeed, the theoretical prediction is limited due to the use of the crystalline NiO model instead of amorphous during DFT calculation. The corresponding statement has been added to the revised manuscript.

The high-angle annular dark-field STEM (HAADF-STEM, Figure 2d) image displays bright spots along with the interfaces of NiO/Ni heterostructure, corresponding to heavy constituent atoms species, which efficiently confirms the immobilization of atomically dispersed Pt atoms in the NiO/Ni nanosheets. The magnified HAADF-STEM image (**Figure 2e**) suggests that the single Pt atoms are exactly immobilized at the interfaces of the NiO/Ni heterostructure. Based on these findings, the atomic environment of Pt atom was explored via the DFT-optimized structure (**Figure 2f-g, Figure S17**), and the result suggests that the Pt atoms are fixed at the Ni positions by binding with O atom and Ni atoms near the interfaces of the NiO/Ni heterostructure. Here, it needs to note that the theoretical prediction is limited due to the use of the crystalline NiO model instead of amorphous during DFT

calculation. To compensate for this limit, an atomic environment of Pt atoms in NiO/Ni, NiO, and Ni supports are further verified by performing X-ray absorption fine structure measurements as shown in **Figure 3** in the revised manuscript. The first-shell EXAFS fitting of Pt_{SA}-NiO/Ni sample (**Figure 3e** and **Table S1**) gives a coordination number (*CN*) of 1.3 for Pt-O contribution and 5.8 for Pt-Ni contribution. Combining the DFT-optimized structure (**Figure S23**), the Pt atoms are mainly immobilized at the interfacial Ni positions by coordinating with one O atom and 5 Ni atoms in Pt_{SA}-NiO/Ni, which is consistent with the conclusion of HAADF-STEM analysis (**Figure 2d-g**).

Q8: "Besides, the fitting curve of Pt XPS spectrums display Pt(IV) species in the samples, which derives from the adsorbed PtCl₆²⁻ ions on the surface of the sample.^{44,45}" on page 8. Pt(IV) also could derive from Pt(IV) oxide as proposed by Birss et al in 1986 (<https://doi.org/10.1149/1.2108978>) and demonstrated by Favaro et al in 2017 (<https://doi.org/10.1039/C7TA00409E>). Can the authors distinguish between these two potential sources of the Pt(IV) XPS signal?

A8: Thanks for your professional question. According to the research reported by Birss et al²⁷, an oxide film grows at Pt substrate under oxygen evolution potential for Pt electrode in alkaline solutions. Further, by employing ambient pressure X-ray photoelectron spectroscopy (APXPS), Favaro et al²⁸ discovered that the Pt oxidation species on the surface of Pt electrode in alkaline solution depend on the applied potential. Especially, only under the high oxygen evolution reaction (OER) potential (+900 mV), the Pt^(IV)O₂ spectral component could be observed as shown in **Figure R21**. From the above discussion, the generation of Pt^(IV)O₂ is mainly determined on the positive OER potential. However, in this study, only the negative HER potential (below 0 V vs RHE) is involved, leading to the quiet difficult generation of Pt^(IV)O₂.

In addition, by X-ray absorption, fine structure measurements shown in **Figure 3**, the first-shell EXAFS fitting of Pt_{SA}-NiO/Ni sample (**Figure 3e** and **Table S1**) gives a coordination number (*CN*) of 1.3 for Pt-O contribution and 5.8 for Pt-Ni contribution. For Pt_{SA}-NiO, the fitting results of EXAFS spectra suggested *CN* about 2.4 for Pt-O

contributions and 2.1 for *CN* for Pt-Ni contributions. Whereas Pt-Ni contribution with 4.9 for *CN* and no Pt-O contributions are found in the fitting of Pt_{SA}-Ni EXAFS spectra. As the contrast, the Pt-O coordination number in PtO₂ is 6 which is extremely large than the *CN* of Pt-O contribution in Pt_{SA}-NiO/Ni, Pt_{SA}-NiO, and Pt_{SA}-Ni,²⁹ further confirming the absence of Pt^(IV)O₂ generated in this work.

Figure R21. Evolution of the surface chemistry studied by operando APXPS, as a function of the applied potential reported by Favaro et al. (a) Pt 4f spectra acquired at 4 KeV as a function of the applied potential (*f*(E)) to the working electrode (from OCP to OER) (OCP: open circuit potential; OER: oxygen evolution reaction; OH_{ads}: adsorbed hydroxyls). (b) Evolution of the surface structure as a function of the applied potential (*f*(E)), from OCP to OER.²⁸

Q9: "Combining the DFT-optimized structure (Figure S18), the Pt atoms are mainly immobilized at the interfacial Ni positions by coordinating with one O atom and five atoms in Pt_{SA}-NiO/Ni, which is consistent with the conclusion of HAADF-STEM analysis (Figure 2d-g)." on page 10. For Pt_{SA}-NiO/Ni, what about the configuration where Pt adsorbs at the bridging site between two surface O? The reported Pt-O *CN* of ~1.3 suggests that Pt could adsorb on both sides of the interface. The authors should consider this.

A9: Thanks for pointing this out. When Pt atom is immobilized at the interfacial Ni position in Pt_{SA}-NiO/Ni, the Pt atom will coordinate with one O atom and five Ni atoms as shown in **Figure R22a**. The first-shell EXAFS fitting of Pt_{SA}-NiO/Ni sample (**Figure 3e** and **Table S1**) gives a coordination number (*CM*) of 1.3 for Pt-O contribution and 5.8 for Pt-Ni contribution, which is consistent with the model of Pt atom immobilized at the interfacial Ni positions in Pt_{SA}-NiO/Ni. However, when the Pt atom is immobilized at the bridging site between the surface O in Pt_{SA}-NiO/Ni, the Pt atom will coordinate with three O atoms and three Ni atoms as shown in **Figure R22b**, which is inconsistent with the result of the first-shell EXAFS fitting of Pt_{SA}-NiO/Ni sample.

In addition, according to the optimized crystal structures deriving from the DFT calculation (**Figure R23**), the formation energy of Pt immobilized at the interfacial Ni positions in Pt_{SA}-NiO/Ni is -2.10 eV, significantly lower than that at the bridging site between the surface O in Pt_{SA}-NiO/Ni (-0.65 eV), suggesting the preference of the interfacial Ni positions occupancy of incorporated Pt in Pt_{SA}-NiO/Ni.

Figure R22. The structure model for the Pt atom immobilized at (a) the interfacial Ni positions and (b) the bridging site between the surface O in Pt_{SA}-NiO/Ni.

Figure R23. The optimized structures and the formation energy of the Pt atom immobilized at the interfacial Ni positions and the bridging site between the surface O in Pt_{SA}-NiO/Ni.

References for this response letter:

- 1 Kothari, V. Polyesters and polyamides. *Elsevier*, 419-440 (2008).
- 2 Lin, C. Polyesters and polyamides, *Elsevier*, 62-96 (2008).
- 3 Wu, L. *et al.* Boosting vanadium flow battery performance by Nitrogen-doped carbon nanospheres electrocatalyst. *Nano Energy* **28**, 19-28 (2016).
- 4 Zhou, K. *et al.* Seamlessly conductive Co(OH)₂ tailored atomically dispersed pt electrocatalyst in hierarchical nanostructure for efficient hydrogen evolution reaction. *Energy Environ. Sci.* **13**, 3082-3092 (2020).
- 5 Zhou, K. *et al.* A Setaria-inflorescence-structured catalyst based on nickel–cobalt wrapped silver nanowire conductive networks for highly efficient hydrogen evolution. *J. Mater. Chem. A* **7**, 26566-26573 (2019).
- 6 Sun, Q., *et al.* Synergistic nanotubular copper-doped nickel catalysts for hydrogen evolution reactions. *Small* **14**, 1704137 (2018).
- 7 Gong, M. *et al.* Nanoscale nickel oxide/nickel heterostructures for active hydrogen evolution electrocatalysis. *Nat. Commun.* **5**, 1-6 (2014).
- 8 Li, X. *et al.* Sequential electrodeposition of bifunctional catalytically active structures in MoO₃/Ni-NiO composite electrocatalysts for selective hydrogen and oxygen evolution. *Adv. Mater.* **32**, 2003414 (2020).
- 9 Zardetto, V. *et al.* Substrates for flexible electronics: A practical investigation on the electrical, film flexibility, optical, temperature, and solvent resistance properties. *J. Polym. Sci., Part B: Polym. Phys.* **49**, 638-648 (2011).
- 10 Fang, S. *et al.* Uncovering near-free platinum single-atom dynamics during electrochemical hydrogen evolution reaction. *Nat. Commun.* **11**, 1029, (2020).
- 11 Ye, S. *et al.* Highly stable single Pt atomic sites anchored on aniline-stacked graphene for hydrogen evolution reaction. *Energy Environ. Sci.* **12**, 1000-1007 (2019).
- 12 Zhang, Y. *et al.* Ultrafine metal nanoparticles/N - doped porous carbon hybrids coated on carbon fibers as flexible and binder-free water splitting catalysts. *Adv.*

- Energy Mater.* **7**, 1700220 (2017).
- 13 Hunt, S. T. *et al.* Activating earth-abundant electrocatalysts for efficient, low-cost hydrogen evolution/oxidation: sub-monolayer platinum coatings on titanium tungsten carbide nanoparticles. *Energy Environ. Sci.* **9**, 3290-3301 (2016).
 - 14 Huang, X. *et al.* High-performance transition metal-doped Pt₃Ni octahedra for oxygen reduction reaction. *Science* **348**, 1230-1234 (2015).
 - 15 Romanchenko, A. *et al.* X-ray Photoelectron Spectroscopy (XPS) study of the products formed on sulfide minerals upon the interaction with aqueous Platinum (IV) chloride complexes. *Minerals* **8**, 578 (2018).
 - 16 Mathew, K. *et al.* Implicit solvation model for density-functional study of nanocrystal surfaces and reaction pathways. *J. Chem. Phys.* **140**, 084106 (2014).
 - 17 Pliego, J. R. *et al.* The cluster-continuum model for the calculation of the solvation free energy of ionic species. *J. Phys. Chem. A* **105**, 7241-7247 (2001).
 - 18 Patel, A. M. *et al.* Theoretical approaches to describing the oxygen reduction reaction activity of single-atom catalysts. *J. Phys. Chem. C* **122**, 29307-29318 (2018).
 - 19 Plimpton, S. Fast parallel algorithms for short-range molecular dynamics. *J. Comput. Phys.* **117**, 1-19 (1995).
 - 20 Laursen, A. *et al.* Nanocrystalline Ni₅P₄: a hydrogen evolution electrocatalyst of exceptional efficiency in both alkaline and acidic media. *Energy Environ. Sci.* **8**, 1027-1034 (2015).
 - 21 Laursen, A. B. *et al.* Climbing the volcano of electrocatalytic activity while avoiding catalyst corrosion: Ni₃P, a hydrogen evolution electrocatalyst stable in both acid and alkali. *ACS Cat.* **8**, 4408-4419 (2018).
 - 22 Lai, J. *et al.* Strongly Coupled nickel-cobalt nitrides/carbon hybrid nanocages with Pt-like activity for hydrogen evolution catalysis. *Adv. Mater.* **31**, 1805541 (2019).
 - 23 Xie, X. *et al.* Electrocatalytic hydrogen evolution in neutral PH solutions: dual-phase synergy. *ACS Cat.* **9**, 8712-8718 (2019).
 - 24 Lv, F. *et al.* Ir-based alloy nanoflowers with optimized hydrogen binding energy

- as bifunctional electrocatalysts for overall water splitting. *Small Methods* **4**, 1900129 (2020).
- 25 Li, Y. *et al.* Ru nanoassembly catalysts for hydrogen evolution and oxidation reactions in electrolytes at various pH values. *Appl. Catal. B: Environ.* **258**, 117952 (2019).
- 26 Hsu, P. C. *et al.* Hydrogen bubbles and the growth morphology of ramified zinc by electrodeposition. *J. Electrochem. Soc.* **155**, D400 (2008).
- 27 Birss, V. I. *et al.* Oxygen evolution at platinum electrodes in alkaline solutions: II. Mechanism of the reaction. *J. Electrochem. Soc.* **133**, 1621 (1986).
- 28 Favaro, M. *et al.* Elucidating the alkaline oxygen evolution reaction mechanism on platinum. *J. Mater. Chem. A* **5**, 11634-11643 (2017).
- 29 Zhang, C. *et al.* Alkali-metal-promoted Pt/TiO₂ opens a more efficient pathway to formaldehyde oxidation at ambient temperatures. *Angew. Chem. Int. Ed.* **51**, 9628-9632 (2012).

REVIEWERS' COMMENTS

Reviewer #1 (Remarks to the Author):

I appreciate the authors consideration of my comments. I read the revised manuscript and the authors response letter, and believe the manuscript is now suitable for publication.

Reviewer #2 (Remarks to the Author):

The authors have revised the manuscript according to my previous comments properly, and now it can be accepted for publication.